mathematical modelling/biomathematics

COVID-19, model with memory, Volterra equations

**Author for correspondence:**
Étienne Pardoux
e-mail: etienne.pardoux@univ-amu.fr

# Estimating the state of the COVID-19 epidemic in France using a model with memory

Raphaël Forien[1], Guodong Pang[2] and Étienne Pardoux[3]

[1]INRAE, BioSP, Centre INRAE PACA, Domaine St-Paul, 84914 Avignon Cedex FRANCE
[2]The Harold and Inge Marcus Department of Industrial and Manufacturing Engineering, College of Engineering, Pennsylvania State University, University Park, PA 16802 USA
[3]Aix–Marseille Université, CNRS, Centrale Marseille, I2M, UMR 7373 13453 Marseille, France

RF, 0000-0002-8901-4921; GP, 0000-0001-6468-0429; ÉP, 0000-0002-2586-4791

In this paper, we use a deterministic epidemic model with memory to estimate the state of the COVID-19 epidemic in France, from early March until mid-December 2020. Our model is in the SEIR class, which means that when a susceptible individual (S) becomes infected, he/she is first exposed (E), i.e. not yet contagious. Then he/she becomes infectious (I) for a certain length of time, during which he/she may infect susceptible individuals around him/her, and finally becomes removed (R), that is, either immune or dead. The specificity of our model is that it assumes a very general probability distribution for the pair of exposed and infectious periods. The law of large numbers limit of such a model is a model with memory (the future evolution of the model depends not only upon its present state, but also upon its past). We present theoretical results linking the (unobserved) parameters of the model to various quantities which are more easily measured during the early stages of an epidemic. We then apply these results to estimate the state of the COVID-19 epidemic in France, using available information on the infection fatality ratio and on the distribution of the exposed and infectious periods. Using the hospital data published daily by Santé Publique France, we gather some information on the delay between infection and hospital admission, intensive care unit (ICU) admission and hospital deaths, and on the proportion of people who have been infected up to the end of 2020.

## 1. Introduction

At the beginning of an epidemic outbreak, some quantities are easier to observe and report than others. For example, the number of hospital deaths related to COVID-19 has been precisely and regularly reported in several countries. Another well-documented quantity is the doubling time of the number of cases (which coincides with that of the number of deaths, as we shall explain). On the other hand, some quantities of interest

are very hard to directly measure or to estimate: the actual number of infected individuals, the true death rate of the disease, and most notably the now famous basic reproduction number $R_0$.

This paper presents a theoretical study of how the 'observable' quantities relate to the 'unobserved' ones, under a very general epidemic model recently developed in [1], and an application of these results to attempt to describe and predict the evolution and current state of the COVID-19 epidemic in France.

The first basic idea is to estimate an exponential growth rate from the available data (the number of hospital admissions or the number of deaths occurring in hospitals), and then to relate it to the parameters of a model of the evolution of the COVID-19 epidemic. Our approach is modelled upon the approach of Tom Britton in two recent preprints [2,3]. Note that the exponential growth rate can be negative, which was the case in particular in France during the two lockdown periods.

The exponential growth rate is the most natural parameter which can be extracted from the data. It equals log(2) divided by the doubling time, i.e. the number of days necessary for the number of daily new cases (or deaths) to double, a notion of which everyone listening to the news at the time of the rise of the epidemic has heard. The other important parameters of the model, including the famous basic reproduction number $R_0$ (the mean number of individuals whom an infectious individual infects before recovering, at the start of the epidemic—that is, while essentially everyone around him/her is susceptible), can be computed from the exponential growth rate and some other parameters of the model.

Our approach has two specific features. First, we shall carefully make explicit which parameters are needed to compute the quantities of interest, and what information can be retrieved from the available data if these parameters are known. The second aspect is that we shall use non-conventional models. The classical SIR and SEIR ODE models are law of large numbers limits of stochastic Markov SIR or SEIR models [4]. For those models to be Markovian, it is necessary that the infectious periods in the case of the SIR model of the various individuals in the population be Independent and identically distributed (i.i.d.) copies of an exponential random variable (respectively, the pair exposed period, infectious period in the SEIR model be i.i.d. copies of a pair of independent exponential random variables). However, in the case of COVID-19, like in most infectious diseases, the assumption that those durations follow an exponential distribution is unrealistic.

Recently, the last two authors of this paper have described the law of large numbers limit model of non-Markovian stochastic epidemic models with arbitrary distribution for the infectious period (respectively, for the pair exposed and infectious period), see [1]. This type of deterministic model is an integral equation of Volterra type, of the same dimension as the classical ODE model. In particular, it is not much more complicated to simulate and compute. The differences between our model and the usual 'Markov' model are not so much in the large time behaviour, but rather in the transient short-term evolution, as was recently observed in [5], which studies a discrete-time model with memory effects in order to account for the inertia of the epidemic and the delays before hospital admission and death. We note that since the start of the epidemic, knowledge about the COVID-19 disease has substantially increased. Our original approach allows us to choose distributions which reflect closely the current knowledge about the disease.

Note that a handful of other works have used similar extended models to analyse the COVID-19 epidemic. In [6], the authors use ODEs with delays, which correspond to our model with deterministic exposed and infectious periods. On the other hand, following the approach initiated by Kermack and McKendrick to analyse the plague epidemic in Mumbai in 1905–1906 [7], [8] uses a transport PDE SEIR model, where the rate of infection by an infectious individual depends upon the time since infection, and the rate at which exposed (respectively, infectious) individuals become infectious (respectively, removed) also depends on the time since infection. The results of the present paper are comparable to those in [8], where the analysis and the treated data concern the various *départements* of Île-de-France, while we compare Île-de-France, Grand Est, Provence–Alpes–Côte d'Azur and Auvergne–Rhône–Alpes. Finally, [9] uses an age-of-infection structure to model the COVID-19 epidemic in France.

Models similar to our non-Markovian SEIR epidemic model appear in several papers (e.g. [10,11]) and in section 4.5 of the recent book [12]. See also [13], which compares an SIR model with delay (which corresponds at the level of the individual-based model to a deterministic infectious period) with the classical SIR ODE model. However, our model seems to be more general than those which appeared earlier, in particular regarding the initial condition and allowing correlated exposed and infectious periods, and as far as we can tell its rigorous interpretation as the functional law of large numbers limit of an individual-based stochastic model established in [1] is new. This interpretation was crucial to suggest the reduction of the dimension of the model which we shall describe below. As a matter of fact, our formula for $R_0$ may be considered as a particular case of a formula which appears on page 141 of [12] (see also [14] and [15]).

The paper is organized as follows. In §2 called 'Methods', we describe our model and the methodology to extract the parameters of our model from the available data. In §3 called 'Results', we describe the results we obtained by applying our method to the COVID-19 epidemic in France. The last section is a discussion of

the conclusions of our work. Finally, an appendix contains the mathematical proof of one crucial result, which relates the exponential growth rate to the various quantities in our model.

# 2. Methods

## 2.1. An SEIR epidemic model with memory for COVID-19

We shall use a deterministic SEIR epidemic model with memory. That model is shown in [1] to be the law of large numbers limit, as the population size tends to infinity, of an individual-based stochastic model, which we first describe.

Assume that each newly infected individual in the population becomes infectious after a random time $\mathcal{E}$ during which he/she is 'exposed' and stays infectious for a random time $\mathcal{I}$, after which he/she does not infect anyone any more, and also cannot be infected (either as a result of acquired immunity, isolation or death). To each individual is associated a copy $(\mathcal{E}_i, \mathcal{I}_i)$ of the random pair $(\mathcal{E}, \mathcal{I})$, in such a way that the $(\mathcal{E}_i, \mathcal{I}_i)_{i \geq 1}$ are independent and identically distributed (abbreviated i.i.d.). The important point which distinguishes our model from most common epidemic models is that we do not assume that the two random variables $\mathcal{E}$ and $\mathcal{I}$ are independent nor that they follow exponential distributions. Hence the stochastic model is not a Markov model, but rather a model with memory, which will also be the case of the limiting model. We assume that infectious individuals attempt to infect other individuals (chosen uniformly from the population) at rate $\lambda(t)$, where $t$ is the current time (the dependence on $t$ of the contact rate $\lambda$ is used to reflect the effect of containment measures such as lockdown, and social behaviour such as use of face masks and physical distancing). As a result, the number of newly infected individuals up to time $t$ can be represented as

$$P\left( N \int_0^t \lambda(s) \overline{S}^N(s) \overline{I}^N(s) \, ds \right),$$

where $P(t)$ is a standard Poisson process, $N$ is the size of the population which is assumed to be fixed throughout the epidemic (the deaths due to the epidemic are counted among the 'removed'), $\overline{S}^N(t)$ (respectively, $\overline{I}^N(t)$) denotes the proportion of susceptible (respectively, infectious) individuals in the population at time $t$. In addition, let $(\overline{E}^N(t), \overline{R}^N(t))$ denote the proportions of exposed and removed individuals at time $t$ in the population, respectively.

The initially exposed or infectious individuals are thought of as having been infected in the past before the initial time $t = 0$. While in the case of exponential exposed and infectious periods, it is quite natural to assume that the remaining exposed (respectively, infectious) periods of the initially exposed (respectively, infectious) individuals have the same law as those of the individuals infected at time $t > 0$ (due to the lack of memory property of the exponential distribution), that is no longer the case in our model.

Therefore, we need to introduce the distribution of the pair $(\mathcal{E}_0, \mathcal{I}_0)$, the remaining exposed and infectious period of an initially exposed individual, as well as the distribution $\mathcal{I}_1$ of the remaining infectious period of an initially infectious individual. More precisely, if $E(0)$ (respectively, $I(0)$) denotes the number of initially exposed (respectively, infectious) individuals, then the collection of random variables $(\mathcal{E}_{0,i}, \mathcal{I}_{0,i})_{1 \leq i \leq E(0)}$, (respectively, $(\mathcal{I}_{1,i})_{1 \leq i \leq I(0)}$) is supposed to be i.i.d. and describes the duration of the remaining exposed and infectious periods of the initially exposed individuals (respectively, the duration of the remaining infectious periods of the initially infectious individuals). We of course assume that the three collections $(\mathcal{E}_{0,i}, \mathcal{I}_{0,i})_{1 \leq i \leq E(0)}$, $(\mathcal{I}_{1,i})_{1 \leq i \leq I(0)}$ and $(\mathcal{E}_i, \mathcal{I}_i)_{i \geq 1}$ are mutually independent, and are globally independent of the above Poisson process $P(t)$.

We further let $S(0)$ (respectively, $R(0)$) denote the number of initially susceptible (respectively, removed) individuals, and define $N = S(0) + E(0) + I(0) + R(0)$, the total population size. For $X = S(0)$, $E(0)$, $I(0)$ or $R(0)$, let $\overline{X} = N^{-1}X$. We assume that as $N \to \infty$,

$$(\overline{S}^N(0), \overline{E}^N(0), \overline{I}^N(0), \overline{R}^N(0)) \to (\overline{S}(0), \overline{E}(0), \overline{I}(0), \overline{R}(0))$$

in probability. Define

$$G^c(t) = \mathbb{P}(t < \mathcal{E}), \quad \Psi(t) = \mathbb{P}(\mathcal{E} \leq t < \mathcal{E} + \mathcal{I}), \quad \Phi(t) = \mathbb{P}(\mathcal{E} + \mathcal{I} \leq t),$$

as well as

$$G_0^c(t) = \mathbb{P}(t < \mathcal{E}_0), \quad \Psi_0(t) = \mathbb{P}(\mathcal{E}_0 \leq t < \mathcal{E}_0 + \mathcal{I}_0), \quad \Phi_0(t) = \mathbb{P}(\mathcal{E}_0 + \mathcal{I}_0 \leq t)$$

and

$$F_1(t) = \mathbb{P}(\mathcal{I}_1 \le t), \quad F_1^c(t) = 1 - F_1(t) = \mathbb{P}(t < \mathcal{I}_1).$$

**Definition 2.1.** Let $\lambda: \mathbb{R}_+ \to \mathbb{R}_+$ be a bounded càdlàg function. The deterministic non-Markovian SEIR model is the solution of the set of integral equations:

$$\overline{S}(t) = \overline{S}(0) - \int_0^t \lambda(s)\overline{S}(s)\overline{I}(s)\,\mathrm{d}s,$$

$$\overline{E}(t) = \overline{E}(0)G_0^c(t) + \int_0^t \lambda(s)G^c(t-s)\overline{S}(s)\overline{I}(s)\,\mathrm{d}s,$$

$$\overline{I}(t) = \overline{I}(0)F_1^c(t) + \overline{E}(0)\Psi_0(t) + \int_0^t \lambda(s)\Psi(t-s)\overline{S}(s)\overline{I}(s)\,\mathrm{d}s,$$

$$\overline{R}(t) = \overline{I}(0)F_1(t) + \overline{E}(0)\Phi_0(t) + \int_0^t \lambda(s)\Phi(t-s)\overline{S}(s)\overline{I}(s)\,\mathrm{d}s.$$

Theorem 3.1 from [1] states that, under a very weak assumption on the joint distribution of $(\mathcal{E}, \mathcal{I})$, the unique solution of the above system of integral equations with memory is the functional law of large numbers limit of the above described non-Markovian model. More precisely, as $N \to \infty$, $(\overline{S}^N(t), \overline{E}^N(t), \overline{I}^N(t), \overline{R}^N(t)) \to (\overline{S}(t), \overline{E}(t), \overline{I}(t), \overline{R}(t))$ in probability, locally uniformly in $t$. We call this model non-Markovian since the stochastic process $(\overline{S}^N(t), \overline{E}^N(t), \overline{I}^N(t), \overline{R}^N(t))$ is not a Markov process in general, because its future evolution depends not only upon its present value but also upon how long the exposed and infectious individuals have already been in the corresponding compartments. For the model to be Markovian, it is necessary that the random variables $\mathcal{E}$ and $\mathcal{I}$ be independent and have exponential distributions. In that case, the integral equation SEIR model reduces to the following ODE model

$$\frac{\mathrm{d}\overline{S}(t)}{\mathrm{d}t} = -\lambda(t)\overline{S}(t)\overline{I}(t), \quad \frac{\mathrm{d}\overline{E}(t)}{\mathrm{d}t} = \lambda(t)\overline{S}(t)\overline{I}(t) - \nu\overline{E}(t)$$

and

$$\frac{\mathrm{d}\overline{I}(t)}{\mathrm{d}t} = \nu\overline{E}(t) - \gamma\overline{I}(t), \quad \frac{\mathrm{d}\overline{R}(t)}{\mathrm{d}t} = \gamma\overline{I}(t),$$

where $\nu$ (respectively, $\gamma$) is the parameter of the exponential law of $\mathcal{E}$ (respectively, $\mathcal{I}$), i.e. $\mathbb{E}(\mathcal{E}) = \nu^{-1}$, $\mathbb{E}(\mathcal{I}) = \gamma^{-1}$. Note that in the Markovian case, the convergence $(\overline{S}^N(t), \overline{E}^N(t), \overline{I}^N(t), \overline{R}^N(t)) \to (\overline{S}(t), \overline{E}(t), \overline{I}(t), \overline{R}(t))$ as $N \to \infty$ is a classical result, see, e.g. [4] for a recent account. The above ODE model is the one which is used by almost all papers dealing with epidemic models, in particular the COVID-19 models, with the exception, to our knowledge, of [6], where the case of fixed $\mathcal{E}$ and $\mathcal{I}$ is considered and [5,8,9]. It is fair to say that the exponential distributions are very poor models for the laws of $\mathcal{E}$ and $\mathcal{I}$, and it may seem strange that the above ODE model is so widely used, while it is based on such unrealistic assumptions. While the large time behaviour of the model (e.g. the endemic equilibrium in the case of a SIS or a SIRS model, see §4.3 in [1]) depends only on the expectation of the random vector $(\mathcal{E}, \mathcal{I})$, and are the same in a Markov and a non-Markov model with identical expected infectious periods, as we shall see below, the transient behaviour of the model depends much more on the details of the law of $(\mathcal{E}, \mathcal{I})$.

Let us now explain the specific model (with two variants) which we will use for the COVID-19 epidemic. In our model, the two random variables $\mathcal{E}$ and $\mathcal{I}$ will be independent. The random variable $\mathcal{E}$ will be distributed on the interval [2, 4]. Concerning the random variable $\mathcal{I}$, we assume a bimodal law with support in $[3, 5] \cup [8, 12]$ (details in §2.4).

The idea behind that type of law for $\mathcal{I}$ is as follows. Our model is close to the SEIRU model of [16], which, thanks to the flexibility of our class of models, we are able to simplify. Indeed, in that paper the individuals are first exposed (state E), then infectious (state I) but without symptoms, and at the end of the I period, a fraction of the individuals are isolated quickly after the onset of symptoms (state R as 'reported'), either because they are admitted to hospital or because they self-isolate at home, while the other individuals are not isolated (state U as 'unreported'), either because they are unable to do so or because they have light or no symptoms. While in the R class, the individuals are isolated and they do not infect susceptibles any more. For that reason, we identify the 'reported' class with part of the 'recovered' class, thus reconciling the two meanings of R. Finally, some of the individuals are infectious when they are in the I class only, while others are infectious in both I and U classes. In our

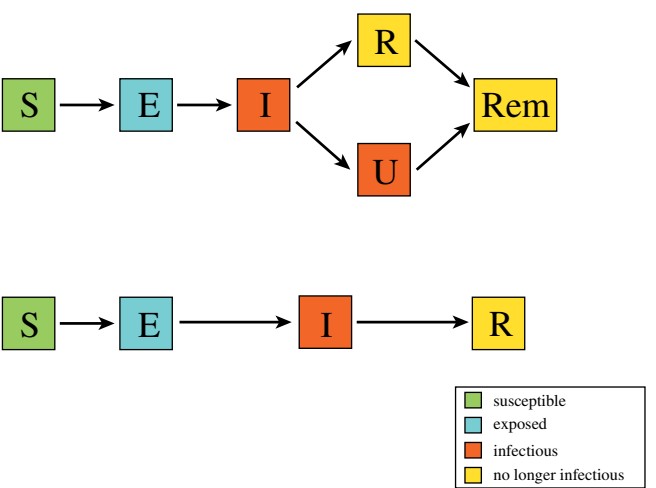

**Figure 1.** Flow chart of the SEIRU model of [16] and the equivalent SEIR non-Markovian model. In the latter, the two infectious compartments (I and U) are merged into one compartment, but the sojourn time $\mathcal{I}$ of the infectious compartment has a bimodal distribution corresponding to the two subpopulations of reported and unreported individuals. Note that R (reported) and Rem (removed) have been merged into a unique compartment.

model, being in the U class is considered as staying longer in the I class (figure 1). This is the motivation for the bimodal distribution of the law of $\mathcal{I}$.

## 2.2. The model with memory during the early phase of the epidemic

During the early phase of the epidemic, the cumulative number of infected individuals remains small compared to the total size of the population. As a result, $\overline{S}(t) \approx 1$ during this phase. Letting $(E(t), I(t), R(t))$ denote the absolute numbers of exposed, infectious and removed individuals during this phase, i.e.

$$(E(t), I(t), R(t)) = (N\overline{E}(t), N\overline{I}(t), N\overline{R}(t)),$$

and assuming that $\lambda$ is constant during this phase, the non-Markovian SEIR model reduces to

$$\left.\begin{aligned}
E(t) &= E(0)G_0^c(t) + \lambda \int_0^t G^c(t-s)I(s)\,\mathrm{d}s, \\
I(t) &= I(0)F_1^c(t) + E(0)\Psi_0(t) + \lambda \int_0^t \Psi(t-s)I(s)\,\mathrm{d}s \\
R(t) &= R(0) + I(0)F_1(t) + E(0)\Phi_0(t) + \lambda \int_0^t \Phi(t-s)I(s)\,\mathrm{d}s.
\end{aligned}\right\} \tag{2.1}$$

and

The initial state of an epidemic is seldom known, and the COVID-19 epidemic is no exception. However, it is always the case that, once sufficiently many individuals have been infected, the cumulative number of infected individuals grows exponentially with some rate $\rho > 0$. We thus look for solutions to

$$\left.\begin{aligned}
E(t) &= \lambda \int_{-\infty}^t G^c(t-s)I(s)\,\mathrm{d}s, \\
I(t) &= \lambda \int_{-\infty}^t \Psi(t-s)I(s)\,\mathrm{d}s, \\
R(t) &= \lambda \int_{-\infty}^t \Phi(t-s)I(s)\,\mathrm{d}s,
\end{aligned}\right\} \tag{2.2}$$

for $t \in \mathbb{R}$ which are of the form

$$E(t) = e\,\mathrm{e}^{\rho t}, \quad I(t) = i\,\mathrm{e}^{\rho t}, \quad R(t) = r\,\mathrm{e}^{\rho t}, \tag{2.3}$$

with $e + i + r = 1$. These equations started from $-\infty$ can be seen as modelling an epidemic which has been growing for a very long time from an infinitesimal proportion of non-susceptible individuals.

One expects (and the results below will confirm that expectation) that $\rho$ is uniquely determined, given $\lambda$ and the law of $(\mathcal{E}, \mathcal{I})$. However, $\lambda$ is not known, while $\rho$ is rather easily estimated from the available data, as we will see below, and we shall conjecture a law for $(\mathcal{E}, \mathcal{I})$, based on the observations of medical doctors (see in particular [17] and also [18]). Hence, we are interested in the inverse problem of expressing the unknown parameter $\lambda$ as a function of $\rho$ and the law of $(\mathcal{E}, \mathcal{I})$, which is the first result of the following Proposition.

**Proposition 2.2.**

(i) *If $\rho > 0$, the system (2.2) admits solutions of the form (2.3) for all $t \in \mathbb{R}$ if*

$$\lambda = \frac{\rho}{\mathbb{E}[e^{-\rho \mathcal{E}}(1 - e^{-\rho \mathcal{I}})]} \tag{2.4}$$

*and with*

$$\boldsymbol{e} = \mathbb{E}[1 - e^{-\rho \mathcal{E}}], \quad \boldsymbol{i} = \mathbb{E}[e^{-\rho \mathcal{E}}(1 - e^{-\rho \mathcal{I}})], \quad \boldsymbol{r} = \mathbb{E}[e^{-\rho(\mathcal{E}+\mathcal{I})}]. \tag{2.5}$$

*Moreover, if $\Theta$ is an independent exponential variable with parameter $\rho$, then (2.3) also solves (2.1) for all $t \geq 0$ with*

$$\left. \begin{array}{ll} G_0^c(t) = \mathbb{P}(\mathcal{E} > t + \Theta \,|\, \Theta < \mathcal{E}), & \Psi_0(t) = \mathbb{P}(\mathcal{E} \leq t + \Theta < \mathcal{E} + \mathcal{I} \,|\, \Theta < \mathcal{E}), \\ \Phi_0(t) = \mathbb{P}(t + \Theta \geq \mathcal{E} + \mathcal{I} \,|\, \Theta < \mathcal{E}), & F_1(t) = \mathbb{P}(t + \Theta \geq \mathcal{E} + \mathcal{I} \,|\, \mathcal{E} \leq \Theta < \mathcal{E} + \mathcal{I}). \end{array} \right\} \tag{2.6}$$

(ii) *If $\rho < 0$, then $(E(t), I(t)) = (\boldsymbol{e} e^{\rho t}, \boldsymbol{i} e^{\rho t})$ solves the first two equations in (2.2) for all $t \in \mathbb{R}$ if $\lambda$ and $\rho$ satisfy (2.4) and if*

$$\boldsymbol{e} = \mathbb{E}[e^{-\rho \mathcal{E}} - 1], \quad \boldsymbol{i} = \mathbb{E}[e^{-\rho \mathcal{E}}(e^{-\rho \mathcal{I}} - 1)].$$

The fact that a solution of the form $R(t) = \boldsymbol{r} e^{\rho t}$ only exists for positive $\rho$ should not come as a surprise, since $t \mapsto R(t)$ is non-decreasing. Also note that, in the case $\rho > 0$, we can rewrite (2.5) with the help of the variable $\Theta$ as follows,

$$\boldsymbol{e} = \mathbb{P}(\Theta < \mathcal{E}), \quad \boldsymbol{i} = \mathbb{P}(\mathcal{E} \leq \Theta < \mathcal{E} + \mathcal{I}), \quad \boldsymbol{r} = \mathbb{P}(\Theta \geq \mathcal{E} + \mathcal{I}). \tag{2.7}$$

The proof of proposition 2.2 is postponed to appendix A.

**Corollary 2.3.** *The basic reproduction number $R_0$, defined as the mean number of secondary infections caused by a single infectious individual in a fully susceptible population, is linked to the initial growth rate of the cumulative number of infected individuals $\rho$ by the relation*

$$R_0 = \lambda \, \mathbb{E}[\mathcal{I}] = \frac{\rho \, \mathbb{E}[\mathcal{I}]}{\mathbb{E}[e^{-\rho \mathcal{E}}(1 - e^{-\rho \mathcal{I}})]}.$$

*This formula remains valid if $\rho < 0$ and if $S(t)$ is still close to 1.*

**Remark 2.4.** Note that if $(\mathcal{E}, \mathcal{I})$ are two independent exponential random variables with parameters $\nu > 0$ and $\gamma > 0$, the equations of definition 2.1 coincide with the law of large numbers limit of the Markovian SEIR epidemic model [4]. In this case, proposition 2.2 agrees with what is already known about Markovian epidemic models. In particular, in the case $\rho > 0$,

$$\lambda = (\rho + \gamma)\left(1 + \frac{\rho}{\nu}\right), \quad \boldsymbol{e} = \frac{\rho}{\nu + \rho}, \quad \boldsymbol{i} = \frac{\rho \nu}{(\rho + \gamma)(\nu + \rho)}, \quad \boldsymbol{r} = \frac{\nu \gamma}{(\rho + \nu)(\rho + \gamma)}.$$

The equivalent formulae for the SIR model (i.e. with $\mathcal{E} = 0$) can be obtained by letting $\nu \to \infty$, in which case we see that $\rho = \gamma(R_0 - 1)$, as in formula (1) in [2].

If, however, we assume that the variables $\mathcal{E}$ and $\mathcal{I}$ are constant and equal to $(t_e, t_i)$, the equations of definition 2.1 can be seen as delay equations. In this case, proposition 2.2 still applies, and the expectations in (2.4) and (2.5) can be omitted, leading to the relation

$$R_0 = \frac{\rho \, t_i}{e^{-\rho t_e}(1 - e^{-\rho t_i})}.$$

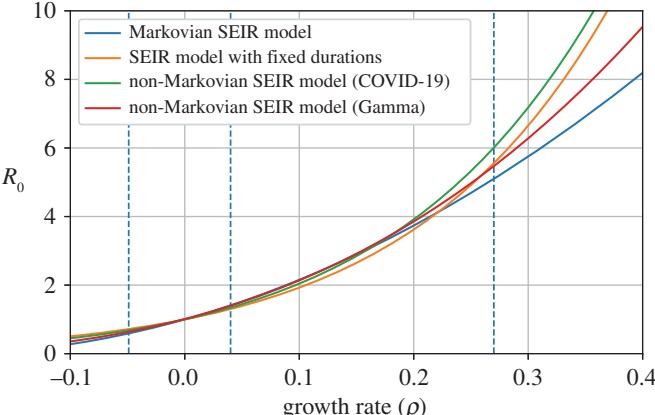

**Figure 2.** Value of $R_0$ as a function of the growth rate $\rho$ for different distributions of the exposed and infectious periods $(\mathcal{E}, \mathcal{I})$ with the same means. Four distributions are displayed: exponential (corresponding to the Markovian SEIR model), fixed, bimodal distribution mimicking COVID-19 (see §2.4) and Gamma distribution. All distributions have a mean exposed time of 4 days and a mean infectious time of 5.4 days, corresponding to a proportion of reported individuals of 0.7. The three dashed vertical lines show the growth rates of the COVID-19 epidemic in mainland France before and during the first lockdown, and during the second wave.

As a result, we see that, for the same growth rate $\rho$, the contact rate and the relative proportions of exposed, infectious and removed individuals vary depending on the distribution of the sojourn times $(\mathcal{E}, \mathcal{I})$, as illustrated in figure 2.

## 2.3. Changing the contact rate during the early phase of the epidemic

In France, as in many countries around the globe, strict lockdown measures were put into place at a fairly early stage of the COVID-19 epidemic. Assuming that, at this stage, the proportion of non-susceptible individuals remains negligible, the non-Markovian model reduces to

$$
\left.\begin{aligned}
E(t) &= E(0)G_0^c(t) + \int_0^t \lambda(s)G^c(t-s)I(s)\,\mathrm{d}s, \\[1mm]
I(t) &= I(0)F_1^c(t) + E(0)\Psi_0(t) + \int_0^t \lambda(s)\Psi(t-s)I(s)\,\mathrm{d}s \\[1mm]
R(t) &= R(0) + I(0)F_1(t) + E(0)\Phi_0(t) + \int_0^t \lambda(s)\Phi(t-s)I(s)\,\mathrm{d}s.
\end{aligned}\right\} \tag{2.8}
$$

and

We shall assume that $E(0)$, $I(0)$ and $R(0)$ are as in (2.5) and that $G_0^c$, $\Psi_0$, $\Phi_0$ and $F_1$ are given by (2.6) with $\Theta$ an exponential random variable of parameter $\rho_0$, and that the function $\lambda(\cdot)$ satisfies

$$
\lambda(t) = \begin{cases} \lambda_0 & \text{if } t \le t_L, \\ \lambda_1 & \text{if } t > t_L, \end{cases}
$$

where $\lambda_i$ satisfies (2.4) with $\rho = \rho_i$ for some $\rho_0 \neq \rho_1$ (note that $\rho_1$ may be negative). Then, by proposition 2.2, for all $t \le t_L$,

$$
E(t) + I(t) + R(t) = \mathrm{e}^{\rho_0 t}.
$$

Moreover, we expect that, after a transitory period $\delta > 0$ following the change of contact rate, the linear system is well approximated by

$$
E(t_L + \delta + t) \approx E(t_L + \delta)\mathrm{e}^{\rho_1 t}, \quad I(t_L + \delta + t) \approx I(t_L + \delta)\mathrm{e}^{\rho_1 t}, \tag{2.9}
$$

for $t \ge 0$ (the interval $\delta$ corresponds to the time it takes for the system to 'forget' the initial growth phase). Summing the three equations in (2.8), we see that,

$$
E(t) + I(t) + R(t) = E(0) + I(0) + R(0) + \int_0^t \lambda(s)I(s)\,\mathrm{d}s.
$$

Let us write $EIR(t) = E(t) + I(t) + R(t)$. As a result, if (2.9) holds, then, for all $t \geq 0$,

$$EIR(t_L + \delta + t) \approx EIR(t_L + \delta) + \lambda_1 \int_0^t I(t_L + \delta) e^{\rho_1 s} \, ds$$

$$\approx EIR(t_L + \delta) + \lambda_1 I(t_L + \delta) \frac{e^{\rho_1 t} - 1}{\rho_1}.$$

Moreover, by linearity, there exist positive constants $C_E(\delta)$, $C_I(\delta)$, $C_R(\delta)$, depending only on $\delta$, $\rho_0$, $\rho_1$ and the distribution of $(\mathcal{E}, \mathcal{I})$, such that

$$E(t_L + \delta) = C_E(\delta) e^{\rho_0 t_L}, \quad I(t_L + \delta) = C_I(\delta) e^{\rho_0 t_L}, \quad R(t_L + \delta) = C_R(\delta) e^{\rho_0 t_L}. \tag{2.10}$$

(We shall omit the dependence on $\delta$ in the following.) These constants are not known explicitly, but can be computed numerically by solving (2.8). Using this and proposition 2.2, we obtain the following

$$EIR(t_L + \delta + t) \approx (C_E + C_I + C_R) \, e^{\rho_0 t_L} + \lambda_1 C_I \, e^{\rho_0 t_L} \frac{e^{\rho_1 t} - 1}{\rho_1}. \tag{2.11}$$

We shall use this later on to estimate the distributions of the delay between infection and hospital admission, hospital death and intensive care unit (ICU) admission.

## 2.4. Distribution of the sojourn times for COVID-19

The evolution of infectivity for COVID-19 from the time of infection remains uncertain, but some early studies already provide constraints on the distribution of the sojourn times $(\mathcal{E}, \mathcal{I})$ for this disease. In particular, [17] estimates that infectiousness starts as early as 2.3 days before the onset of symptoms and declines within 7–8 days of symptom onset (see also [18]). Assuming that symptoms start on average 5.2 days after infection (as reported in [17]), we conclude that $\mathcal{E}$ is between 2 and 4 days, and that $\mathcal{I}$ is at least 3 days (if the infected individual is isolated shortly after the onset of symptoms) and no more than 10 to 12 days.

For the exposed time $\mathcal{E}$, we shall take a linear combination of the form $2 + 2 X_1$, where $X_1$ is a symmetric Beta random variable with support in [0, 1]. We then assume that infected individuals are divided in two groups. Individuals in the first group, called the *reported* individuals, are isolated shortly after the onset of symptoms, either because they self-isolate at home or because they are admitted to a hospital, where we assume that they do not infect anyone else. For these individuals, we assume that $\mathcal{I} = 3 + X_2$, where $X_2$ is a symmetric Beta random variable with support in [0, 1]. By contrast, individuals in the second group remain infectious for much longer, either because they show very mild symptoms or no symptoms at all or because they fail to be isolated. For these *unreported* individuals, we assume that $\mathcal{I} = 8 + 4 X_3$, where $X_3$ is a symmetric Beta random variable with support in [0, 1].

Let $p_R$ be the proportion of reported individuals and let $Y$ be a Bernoulli random variable with parameter $p_R$, independent of $X_2$ and $X_3$. Then we can write

$$\mathcal{I} = Y(3 + X_2) + (1 - Y)(8 + 4 X_3).$$

We thus see that the distribution of $\mathcal{I}$ is bimodal, with a first peak at around 3.5 days and a second one around 10 days.

It seems that asymptomatic individuals are less infectious than symptomatic ones (see [19]). We could have included in the model different values of infectivity depending upon the duration of the infectious period of each individual, using the theory developed in [20], but we lack quantitative information about the various levels of infectivity to make serious predictions, while the available data do not allow us to estimate many parameters. Note also that, as shown in [20], during the early phase of the epidemic, the product of the number of asymptomatic patients and their infectivity determines the evolution of the epidemic. Overestimating the infectivity leads to an underestimation of the number of cases, which has no significant impact on the dynamics of the epidemic during the early phase, but may lead to underestimation of the proportion of the population which has acquired immunity at the end of this early phase.

## 2.5. Estimating the contact rate prior to lockdown measures

In France, the COVID-19 epidemic started with a rapid exponential growth (of the number of cases, number of hospitalized patients, number of deaths), followed by a slowing down of the epidemic due

to the lockdown measures put into place around 16 March [21], settling after a few weeks to an exponential decrease of the number of newly hospitalized patients and new deaths. This decrease continued after the partial lifting of travel restrictions on 11 May, but came to a halt at the beginning of the summer, before hospital admissions, hospital deaths and ICU admissions started to increase again during the summer. This growth was stopped in mid-November, a few weeks after the second lockdown was put into place.

Since testing has been limited to a subset of symptomatic individuals and tests were performed at various intervals following symptom onset and with unequal intensity, the number of reported positive cases might not be exactly proportional to the true number of infected individuals throughout the different stages of the epidemic (and more importantly the ratio of infected individuals to tested individuals may vary significantly between the different phases). Hospital deaths, however, have been reported daily from 15 February. Moreover, from 18 March onwards, Santé Publique France has published daily reports of newly hospitalized patients, new admissions in ICU and new deaths in each administrative *département* [22].

Assuming that the distributions of the intervals between infection and hospital admission, admission in ICU and death do not vary over time, the growth rate of these quantities is necessarily identical to that of the cumulative number of infected individuals (see equation (2.13) below). As a result, if the distribution of the exposed and infectious periods $(\mathcal{E}, \mathcal{I})$ is known, we can infer the contact rate $\lambda$ corresponding to a given growth rate $\rho$ using (2.4).

Note that, since Santé Publique France only started publishing regional data on 18 March, the growth rate for the initial phase in each region is inferred from the slope of the cumulative deaths during the first week of lockdown. Since the delay between infection and death in hospital is believed to be at least 10 days (e.g. [23]), it is safe to assume that all the individuals who died during this period were infected before lockdown measures took effect.

If, moreover, one knows the distribution of the interval $\mathcal{D}$ between infection and death, for example, as well as the infection fatality ratio $f$ (the probability that an individual eventually dies, given that he or she has been infected), it is possible to infer the state of the epidemic. Indeed, the cumulative number of deaths at time $t \geq 0$, denoted by $\Lambda_D(t)$, is then given by

$$\Lambda_D(t) = fN \int_0^t \left( -\frac{d\overline{S}(s)}{ds} \right) \mathbb{P}(\mathcal{D} < t - s) \, ds = fN \, \mathbb{E}[1 - \overline{S}(t - \mathcal{D})], \tag{2.12}$$

where $N$ is the total population size (the expectation is taken with respect to the random variable $\mathcal{D}$).

Note that since deceased individuals are no longer infectious, $\mathcal{D}$ should be coupled with $(\mathcal{E}, \mathcal{I})$ so that, on the event that an individual dies, $\mathcal{E} + \mathcal{I} < \mathcal{D}$ (and similarly for individuals who are admitted to a hospital). As a result, individuals who die at time $t$ (and who were infected at some time $t - \mathcal{D}$), should all be in the removed (R) state. The fact that $\overline{S}(t - \mathcal{D})$ appears in (2.12) instead of $\overline{R}(t - \mathcal{D})$ simply comes from the fact that $\mathcal{D}$ denotes the delay between the time at which an individual *becomes infected*, and the time at which he/she dies.

Recall that, during the early phase of the epidemic,

$$N(1 - \overline{S}(t)) \approx E(t) + I(t) + R(t) = e^{\rho_0 t},$$

where $t$ is the time elapsed since the (unknown) time of the start of the epidemic (i.e. the theoretical time at which only one individual was infected in the population), and $E(t)$, $I(t)$ and $R(t)$ solve the linearized system (2.1). Thus, during this phase,

$$\Lambda_D(t) = f \, \mathbb{E}[\mathbf{1}_{\{\mathcal{D} \leq t\}} \, e^{-\rho_0 \mathcal{D}}] e^{\rho_0 t}. \tag{2.13}$$

It is thus possible to infer $\rho_0$ from the slope of $t \mapsto \log(\Lambda_D(t))$ during this initial phase and, using (2.4), one can then deduce the value of the contact rate before lockdown measures, $\lambda_0$ (given that the distribution of $(\mathcal{E}, \mathcal{I})$ is known).

In order to estimate the start date of the epidemic and to be able to solve the system (2.1), we need information on $f$ and $\mathcal{D}$. We shall see in §2.7 how to obtain information on the distribution of $\mathcal{D}$ using what happens after a change in the contact rate.

## 2.6. Estimating the contact rate during lockdown measures

At the start of lockdown measures, the contact rate in the population dropped sharply over the course of a few days. As a result, the daily number of deaths in hospitals started to increase more slowly, before

decreasing at a steady rate after approximately three weeks. In addition, the number of new hospital admissions and ICU admissions also decreased at a steady rate after a (shorter) transition period.

Assuming that, at this point, the proportion of susceptible individuals in the overall population remains close to 1, this corresponds to the situation described by (2.8) with $\rho_1 < 0$. Using (2.11) and (2.12) and choosing $t$ large enough (such that $\mathbb{P}(\mathcal{D} > t)$ is small), we have

$$\Lambda_D(t_L + \delta + t) \approx f(C_E + C_I + C_R)\, e^{\rho_0 t_L} + \frac{f \lambda_1 C_I\, e^{\rho_0 t_L}}{\rho_1}(\mathbb{E}[e^{-\rho_1 \mathcal{D}}]e^{\rho_1 t} - 1). \tag{2.14}$$

In particular, there exists a constant $C > 0$ such that the time derivative of the above reads

$$\Lambda_D'(t_L + \delta + t) = C\, e^{\rho_1 t},$$

and the same holds for hospital admissions (with a different constant $C$). As a result, we can estimate $\rho_1$ from the cumulative number of hospital admissions and hospital deaths, and deduce the value of the contact rate during lockdown using (2.4).

Here and in what follows, we assume that the distribution of the sojourn times $(\mathcal{E}, \mathcal{I})$ is unaffected by the various containment measures put into place during the course of the epidemic. This may not be true, especially if contact tracing and massive testing are implemented at a sufficient scale in the population, but the extent of such measures in France, at least during the early stages of the COVID-19 epidemic, remained very limited.

## 2.7. Estimating the distribution of the delay between infection and hospital death or hospital admission

We have seen above that (2.14) can be used to determine the contact rate during lockdown measure, but in fact we are able to estimate all the constants appearing in (2.14) by fitting an exponential curve to the cumulative number of hospital deaths. More precisely, we can estimate the values of both

$$A = \frac{f \lambda_1 C_I\, e^{\rho_0 t_L}}{\rho_1} \mathbb{E}[e^{-\rho_1 \mathcal{D}}], \quad \text{and} \quad B = f(C_E + C_I + C_R)\, e^{\rho_0 t_L} - \frac{f \lambda_1 C_I\, e^{\rho_0 t_L}}{\rho_1}.$$

Adding relation (2.13) with $t = t_L$ (and assuming that $\mathcal{D} < t_L$ almost surely), we also have

$$\Lambda_D(t_L) = f\, \mathbb{E}[e^{-\rho_0 \mathcal{D}}]\, e^{\rho_0 t_L}. \tag{2.15}$$

In addition to $f$, the only unknowns here are $\mathbb{E}[e^{-\rho_0 \mathcal{D}}]$, $\mathbb{E}[e^{-\rho_1 \mathcal{D}}]$ and $t_L$. Substituting $f\, e^{\rho_0 t_L} = \Lambda_D(t_L)\mathbb{E}[e^{-\rho_0 \mathcal{D}}]^{-1}$ in the first two equations, we obtain

$$\left.\begin{aligned} \mathbb{E}[e^{-\rho_0 \mathcal{D}}] &= \frac{\Lambda_D(t_L)}{B}\left[(C_E + C_I + C_R) - \frac{\lambda_1 C_I}{\rho_L}\right] \\[2mm] \mathbb{E}[e^{-\rho_1 \mathcal{D}}] &= \frac{A}{B}\left[\frac{\rho_1}{\lambda_1 C_I}(C_E + C_I + C_R) - 1\right]. \end{aligned}\right\} \tag{2.16}$$

and

These two identities do not fully characterize the distribution of $\mathcal{D}$, but if we assume that this distribution belongs to some two-parameter family, it is possible to estimate the corresponding parameters. In the following, we shall assume that $\mathcal{D}$ follows a Gamma distribution with parameters $(\theta, k)$. Then, for all $\rho > -\theta^{-1}$,

$$\mathbb{E}[e^{-\rho \mathcal{D}}] = (1 + \rho\, \theta)^{-k}.$$

The parameters $\theta$ and $k$ can then be computed numerically by inverting (2.16). Then, using (2.15), we can deduce the value of $t_L$, the time elapsed since the beginning of the epidemic at the start of lockdown. We can further note that one does not need to know the value of $f$ in order to compute (2.16), and to estimate the distribution of $\mathcal{D}$. This is fortunate, as the parameter $f$ is difficult to estimate precisely.

Note that the Gamma distribution has support in $[0, \infty)$ and so does not satisfy the requirement that $(\mathcal{E}, \mathcal{I})$ and $\mathcal{D}$ be coupled so that, on the event that an individual dies, $\mathcal{E} + \mathcal{I} < \mathcal{D}$. However, we shall see below that the average delay between infection and hospital admission (hence hospital deaths) is large enough that this inequality can be achieved with high probability. Given the low percentage of individuals who are admitted to hospitals (of the order of 2–3%), this does not significantly affect our results.

We proceed in the same way for the delay between infection and hospital admission and ICU admission. Once we have estimated the parameters of the delay distribution for each of these events, we can compute the probability of hospital admission and ICU admission as follows.

## 2.8. Estimating the probability of hospital and ICU admission

Note that if $p_H$ is the probability that a given infected individual is admitted to a hospital at some point, and if $\mathcal{D}_H$ is the interval between the time of infection and hospital admission, then, the cumulative number of hospitalized patients at time $t \geq 0$, denoted by $\Lambda_H(t)$, is given by

$$\Lambda_H(t) = p_H N \mathbb{E}[1 - \overline{S}(t - \mathcal{D}_H)].$$

Hence, during the early phase of the epidemic,

$$\Lambda_H(t) = p_H \mathbb{E}[\mathbf{1}_{\{\mathcal{D}_H \leq t\}} \, e^{-\rho_0 \mathcal{D}_H}] e^{\rho_0 t}.$$

As a result, one can obtain the probability of being admitted to hospital as a function of $f$ through the relation

$$p_H = f \frac{\Lambda_H(t) \mathbb{E}[e^{-\rho_0 \mathcal{D}}]}{\Lambda_D(t) \mathbb{E}[e^{-\rho_0 \mathcal{D}_H}]}, \tag{2.17}$$

for any time $t \in [0, t_L]$ for which $\mathcal{D} < t$ and $\mathcal{D}_H < t$ almost surely, where $t_L$ is the time at which lockdown measures are put into place.

## 2.9. Estimating the contact rate after the easing of the first lockdown measures

The easing of lockdown restrictions was organized in different phases. On 11 May, people were allowed to leave their homes, shops started to reopen and schools progressively welcomed pupils again. On 2 June, bars and restaurants reopened in most of the country, and on 22 June, all schools reopened, along with cinemas, and most activities resumed, although sanitary measures continued to be enforced (e.g. wearing a mask remained mandatory in many public places including trains and public transport).

As we shall see below, at the end of the first lockdown period, the proportion of susceptible individuals in the population had already dropped by 5 to 10%, at least in Île-de-France and the Grand Est and Hauts-de-France regions. As a result, the approximation $\overline{S}(t) \approx 1$ may not be valid at this point, preventing us from directly applying (2.4). However, since the daily new infections remain at low levels over short time periods, the value of $\overline{S}(t)$ does not vary much during the period over which we estimate the new growth rates $\rho_2$, $\rho_3$, $\rho_4$. It follows that we can replace $\lambda$ by the effective rate $\lambda_e = \overline{S}(t_E)\lambda$ in (2.4), where $\overline{S}(t_E)$ is the proportion of susceptible individuals at the end of the lockdown period. We hence deduce the corrected value of the effective contact rate during each period following the easing of lockdown restrictions. These effective rates correspond to effective reproduction numbers $R_e = \overline{S}(t_E)R_0$, which reduce the original 'basic reproduction number' $R_0$.

## 2.10. Estimating the state of the epidemic

Once we have estimated the contact rates prior to lockdown ($\lambda_0$), during lockdown ($\lambda_1$) and after lockdown ($\lambda_i$, $i \geq 2$), as well as the interval between the (theoretical) index case and the start of lockdown $t_L$, we are in a position to compute the state of the epidemic. To do this, we numerically solve the equations of definition 2.1 with

$$\lambda(s) = \begin{cases} \lambda_0 & \text{if } s < t_L, \\ \lambda_1 & \text{if } t_L \leq s < t_E, \\ \lambda_i & \text{if } t_i \leq s \leq t_{i+1}, \text{ for } i \geq 2, \end{cases}$$

where $t_L$ is the time at which lockdown measures are implemented and $t_E$ is the time at which these measures are eased. We also take $t_2 = t_E$, and $t_i$ for $i \geq 3$ denotes the subsequent times of changes of the contact rate in the population. For the initial condition, using proposition 2.2, we choose

$$\overline{S}(0) = 1 - \frac{1}{N}, \quad \overline{E}(0) = \frac{e}{N}, \quad \overline{I}(0) = \frac{i}{N}, \quad \overline{R}(0) = \frac{r}{N},$$

with $e$, $i$, $r$ as in (2.1) and $G_0^c$, $\Psi_0$, $\Phi_0$, $F_1$ as in (2.6), choosing $\rho$ equal to $\rho_0$, the growth rate prior to lockdown in (2.1) and as the parameter of the exponential random variable $\Theta$ in (2.6). In this way, by proposition 2.2, $1 - \overline{S}(t) = (1/N)e^{\rho_0 t}$ for all $t \leq t_L$, as expected.

It might seem counterintuitive to start the epidemic with some fraction of removed individuals $(\overline{R}(0) > 0)$, even more so as we assume that only one individual is not susceptible at this time. One should keep in mind that we are not trying to estimate the *true* initial state of the epidemic; we merely find a suitable initial condition so that the observed exponential growth prior to lockdown measures fits the observed data. Starting our model with exactly one infectious individual at time 0 would lead to the same exponential growth behaviour, but after a short transitory period which would shift our predictions.

# 3. Results

We present the results of our estimations in four different French regions: Île-de-France, Grand Est, Provence–Alpes–Côte d'Azur (PACA) and Auvergne–Rhône–Alpes. The epidemic was first detected in the Grand Est region, after which it quickly spread to the Paris region, and then the whole country. As a result, some regions were much more affected than others during the first wave in March and April, and this can be seen directly in the hospital data from Santé Publique France [22].

## 3.1. Growth rates of the epidemic

We measured the growth rate of the cumulative number of infected individuals during the early phase of the epidemic by fitting an exponential curve to the cumulative number of deaths in each patch, between 19 and 26 March (earlier data were only available at the national level). Despite the fact that very strict lockdown measures were already effective at the time, hospital deaths continued to increase exponentially until at least 26 March, mainly due to the fact that infected individuals who died of COVID-19 during this period had been infected before national lockdown, hence during the exponential growth phase.

We can see that the initial growth rate in the Grand Est region is the slowest, despite the fact that this is where the epidemic first started. This can be attributed to the fact that some public gatherings were banned in this region about 10 days before the national lockdown. Hence this might not reflect the growth rate of the epidemic from its very start, which might be closer to the one measured at the national level (doubling time of 2.5 days, using hospital deaths data from 1 March).

After a few weeks of lockdown, daily hospital admissions, hospital deaths and ICU admissions started to decline, again exponentially, at a steady rate in each region. We observe a drop in the number of daily admissions and deaths every weekend, which was systematically compensated at the start of the following week, indicating reporting delays. We thus fitted *simultaneously* three exponential curves to the *cumulative* number of hospital admissions, hospital deaths and ICU admissions between 3 April and 11 May (starting only on 13 April for ICU admissions and deaths). The obtained values are given in the second line of table 1. Note that the halving time during lockdown was much longer that the doubling time prior to lockdown, explaining why it had to remain in place for several months in order to significantly reduce the number of infectious individuals.

Using the same method, we estimated the growth rate of the cumulative number of infected individuals in subsequent stages of the epidemic (figure 3). The growth rate of the epidemic remained negative until June, and hospital admissions, followed by ICU admissions and deaths, started to increase again during the summer. This increase continued during the fall in every region, but much more slowly than before the first lockdown, probably due to all the measures put into place since then (mandatory masks in many public places, more people working from home, etc.). We can also note that the growth rate varies between regions, and that it is higher in regions that were less hit by the first wave. This can be explained by the fact that, since fewer people have been infected during the first wave in such places, the overall immunity level is also lower, and hence the epidemic spreads faster. The second wave was then stopped after the start of the second national lockdown on 30 October. The latest data also show a plateau, and even a slight increase in some regions of the hospital admissions over the last weeks of December. If confirmed, this trend would mean that a third wave might hit again very early in 2021.

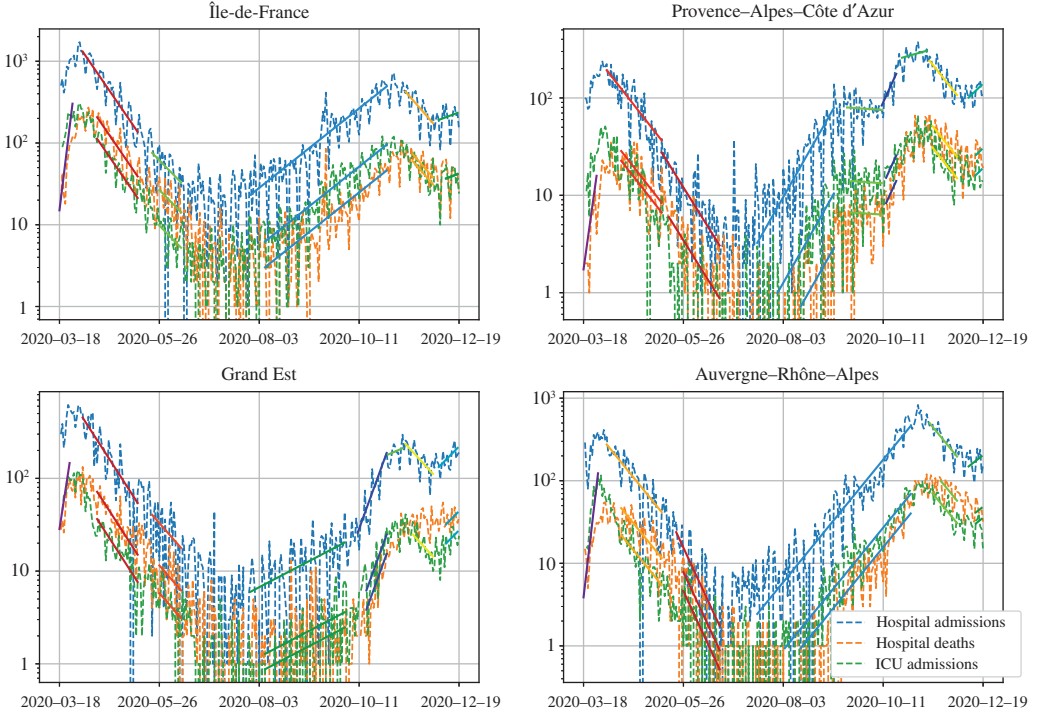

**Figure 3.** Growth rates of the cumulative number of infected individuals before, during and after lockdown in France, deduced from the number of hospital admissions, ICU admissions and hospital deaths in each patch. In each instance, an exponential curve was fitted simultaneously to each cumulative number of hospital admissions, ICU admissions and deaths.

**Table 1.** Doubling and halving times measured during the main phases of the epidemic in each of the regions are presented here. Doubling times prior to lockdown are measured using hospital deaths between 19 and 26 March, but are consistent with what is obtained for the whole country (2.5 days) using data from 1 March.

| | region | | | |
| --- | --- | --- | --- | --- |
| | Île-de-France | Grand Est | PACA | Auvergne–Rhône–Alpes |
| doubling time before lockdown (from hospital deaths) | 2.1 days | 3 days | 2.8 days | 2 days |
| halving time during first lockdown | 11.6 days | 12.4 days | 15.9 days | 13.9 days |
| doubling time during the second wave | 21.4 days | 36.9, then 7 days | 11.5, then 9.2 days | 14 days |
| halving time during second lockdown | 14.9 days | 17.7 days | 16.2 days | 14.1 days |

## 3.2. Reproduction numbers and the effect of lockdown measures

Using the relation (2.4) and our assumptions on the distribution of $(\mathcal{E}, \mathcal{I})$ from §2.4, we can estimate the values of the reproduction number $R_0$ in each region during each phase of the epidemic. Since the growth rates prior to lockdown and during lockdown are relatively uniform across the different regions, we use the same value for all patches. For the second wave, we compute $R_0$ with the two extreme values (doubling every 11.5 days or every 21.4 days).

The proportion of reported individuals affects both the expectation of the infectious period $\mathbb{E}[\mathcal{I}]$ and the whole distribution of $(\mathcal{E}, \mathcal{I})$. Considering corollary 2.3, we thus observe that the estimated value of the reproduction number depends on the proportion of reported individuals, and more generally on the whole distribution of $(\mathcal{E}, \mathcal{I})$, as shown by table 2.

The relatively large values (around 6) obtained for small proportions of reported individuals suggest that this proportion is closer to 0.8 than to 0.2. Moreover, the values we obtain, even for large proportions

**Table 2.** Values of $R_0$ during the three main phases of the epidemic for different proportions of reported individuals, using the sojourn time distribution from §2.4.

| proportion of reported individuals | $R_0$ prior to lockdown | $R_0$ during the first lockdown | $R_0$ during the second wave |
|---|---|---|---|
| 0 | 6.3 | 0.66 | 1.28–1.56 |
| 0.2 | 6 | 0.67 | 1.27–1.55 |
| 0.8 | 4.2 | 0.73 | 1.21–1.42 |
| 1 | 3.4 | 0.79 | 1.16–1.32 |

**Table 3.** Estimates of the reproduction number in the literature

| country | $R_0$ prior to lockdown | $R_0$ during lockdown | accounting for exposed period | reference |
|---|---|---|---|---|
| France | 2.9–3.35 | 0.55–0.72 | implicit | [5] |
| France | 3.1–3.3 | 0.47 | no | [24] |
| France | 3.3 | 0.5 | yes | [23] |
| France | 3.4 ± 0.1 | 0.65 ± 0.04 | yes | [25] |
| China | 4.7–6.6 | / | yes | [26] |

of reported individuals, are sometimes larger than other estimates in the literature (table 3), although we note that our estimate of $R_0$ fits well the results in [26]. We can attribute this to two things. First, as shown in figure 2, the Markovian SEIR model tends to slightly underestimate $R_0$ in the two regimes we are interested in (growth in the early phase and decrease under lockdown). Second, some studies do not take into account the exposed period $\mathcal{E}$ in their models. However, corollary 2.3 shows that neglecting the exposed period leads to underestimating of the reproduction number $R_0$, here by a factor of 0.76 for each day of exposed period ($e^{-\rho} = 0.76$ with $\rho = 0.27$). Finally, uncertainty about the infectious period $\mathcal{I}$ also affects the estimates of the reproduction number. More details are given on this in appendix D, using a sensitivity analysis.

## 3.3. State of the epidemic and acquired immunity at the population level

Using (2.13) and §2.10, we can estimate the state of the epidemic in each region, given the infection fatality ratio $f$ and the distribution of the delay $\mathcal{D}$. Unfortunately, these two quantities are notoriously hard to measure during the early stage of the epidemic.

The infection fatality ratio for the COVID-19 epidemic in France has been estimated in at least two studies [23,24]. Both studies found a fatality ratio of 0.5%, with significant variation across age classes in [23]. These studies only account for hospital deaths, as we do, even though a significant number of deaths take place outside hospitals, mainly in nursing homes. Hence this ratio has to be corrected (roughly by a factor 1.6 [24]) to obtain the true infection fatality ratio of COVID-19 (and to obtain correct predictions for the expected number of deaths). Nevertheless, since we use hospital deaths to calibrate our SEIR model, we shall use the infection fatality ratio estimated by [23,24] when using (2.13).

We can at least bound this ratio from below using the observed excess mortality in some regions. For example, in Lombardy (Italy), 16 973 people died of COVID-19, for a total population of 10 million, showing that this ratio is at least 0.17% (or 0.11% for hospital deaths). Another estimate of this ratio was obtained in a small town in Germany, where around 1000 individuals were tested, out of which 15.5% tested positive and seven people died, yielding an infection fatality ratio of 0.37% [27] (corresponding to 0.23% for hospital deaths).

On the other hand, the infection fatality ratio can be bounded from above by the apparent death rate, that is to say, the ratio between COVID-19-related deaths and declared positive cases, at least while the epidemic seems to be receding, as is the case in France in June 2020, with less than 50 deaths per day. Taking only hospital deaths into account, this suggests that $f$ is no more than 12%. Other countries have much lower apparent death rates, e.g. South Korea (2.3%), Germany (4.6%). These discrepancies

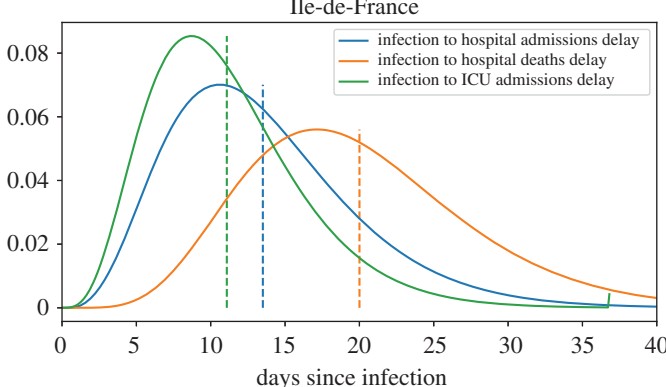

**Figure 4.** Probability density functions of the delays between infection and hospital admission (blue), ICU admission (green) and Hospital death (orange) estimated in Île-de-France. Means are indicated by dashed vertical lines.

**Table 4.** Mean and standard deviations of the delays between infection and hospital admission, ICU admission and hospital deaths obtained from (2.16).

| region | hospital admission | ICU admission | hospital death |
|---|---|---|---|
| Île-de-France | 13.6 ± 6.3 days | 11.1 ± 5.2 days | 20.1 ± 7.6 days |
| Grand Est | 11.7 ± 5.5 days | 9.3 ± 2.6 days | 16.2 ± 7.5 days |
| PACA | 12.7 ± 6.7 days | 13.9 ± 5.3 | 23.7 ± 9 days |
| Auvergne–Rhône–Alpes | 11.4 ± 4 days | 8.8 ± 1.7 days | 20 ± 8.4 days |

can mostly be attributed to differences in testing capacities and to a lesser extent to differences in hospital capacities.

Using the method described in §2.7, we estimated the parameters $(k, \theta)$ for the delay between infection and each of the three possible events (hospital admission, ICU admission and hospital death). The means and standard deviations of the distributions we obtained are given in table 4 and the distributions obtained in Île-de-France are shown on figure 4.

The delays appear to be slightly shorter in the Grand Est region than everywhere else. This can be a consequence of the fact that the growth rate measured during the first weeks of lockdown does not reflect the true initial growth rate of the epidemic in this region, as noted in §3.1.

We estimated the probability of being admitted to hospital given infection. This obviously depends on the infection fatality ratio, but, on the other hand, the death to admission ratio (i.e. the probability of dying given that one is admitted to a hospital) is constant. We estimated that the death to admission ratio was 0.19 in Île-de-France, 0.20 in the Grand Est region, 0.14 in PACA and 0.17 in Auvergne–Rhône–Alpes. Note that we consider that these are constant throughout the epidemic, which may not be the case, especially if hospitals become overwhelmed by the influx of patients. This could for example explain the discrepancy between the death to admission ratios in the different regions, which are computed using the data from the first weeks of lockdown.

Using the resulting estimates on the delay distributions and the probabilities of hospital admission and ICU admission, we can simulate the deterministic SEIR model from definition 2.1 with the contact rates estimated in figure 3, and compute the number of hospital admissions, ICU admissions and hospital deaths in the model. The result is represented in figure 5. The fact that our delay distributions satisfy (2.16) ensures that our model reproduces the initial growth before 16 March and the exponential decrease during the first lockdown.

Interestingly, in two regions (PACA and Grand Est), we are not able to fit the decrease during the second lockdown without an intermediary step with a slower increase around the time at which the lockdown started (figure 8). This can be explained by the fact that, two weeks before the second lockdown started, many large cities declared a nightly curfew from 21.00 to 6.00.

Figure 6 shows the predicted levels of immunity (i.e. the proportion of infected individuals) up to 31 December in Île-de-France for three values of the infection fatality ratio $f$, using the delay distributions estimated above. As expected, higher values of this ratio lead to lower predicted levels of immunity.

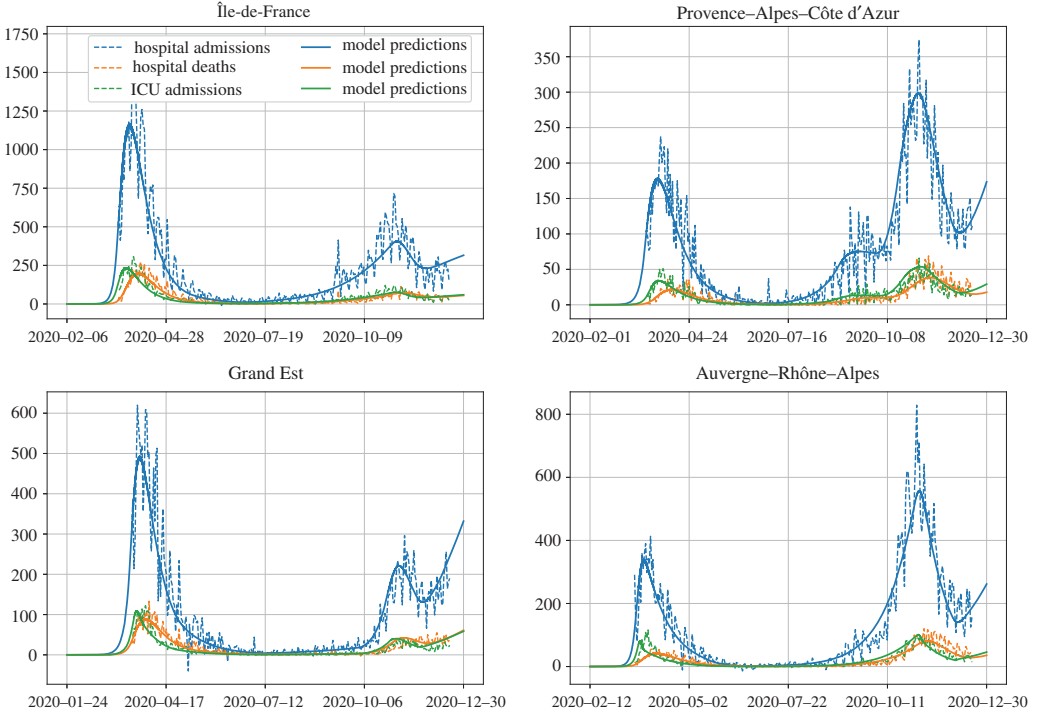

**Figure 5.** Model predictions and hospital data throughout the epidemic in four regions. In each region, we confront the daily hospital admissions, ICU admissions and hospital deaths due to COVID-19 to the predictions of our model. The model starts with a single infected individual at a different date in each region, and ends on 31 December. The value used for the infection fatality ratio $f$ was 0.5%, following [24], and the proportion of reported individuals was 0.8.

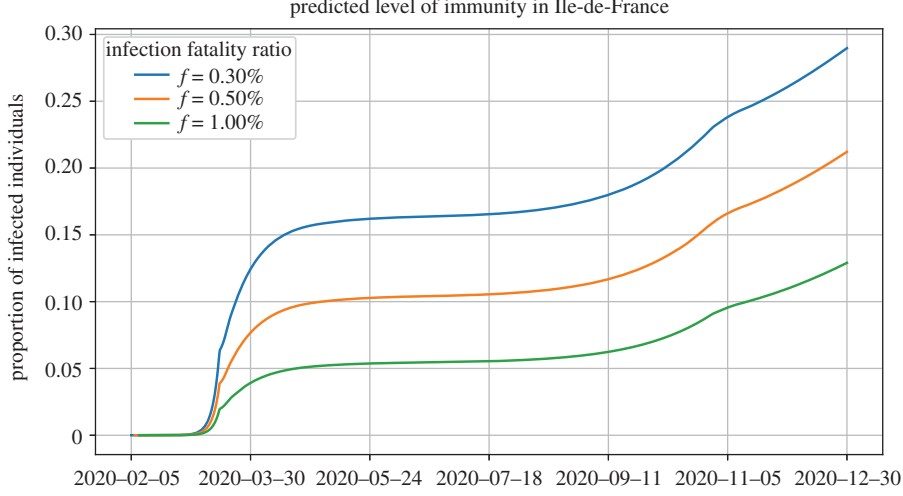

**Figure 6.** Predicted level of immunity in Île-de-France as a function of the infection fatality ratio $f$. All the simulations use $p_R = 0.8$. As expected, larger infection fatality ratios lead to lower predicted immunity, since fewer people need to have been infected to reproduce the observed number of deaths.

Note that underestimating $\mathcal{D}$ has the same effect as overestimating the infection fatality ratio, as can be noted from (2.13). As a result, it is equally important to estimate $f$ and the distribution of $\mathcal{D}$, a fact which is too easily overlooked.

For the fatality ratio estimated by [23,24] ($f = 0.5\%$), we estimate that around 20% of the population has been infected by the end of December in Île-de-France. In the Grand Est region, this proportion is 25%, while in PACA, only 15% seem to have been infected, and 17% in Auvergne–Rhône–Alpes. As expected, regions that were infected later during the first wave now have lower levels of immunity than the two regions where the epidemic started early. As a consequence, during a possible third

epidemic wave, regions with relatively lower cumulative number of death are more at risk of seeing a sharper increase in the number of cases.

# 4. Discussion

One of our goals in this article was to study, in the context of an ongoing epidemic, which knowledge about the parameters of a given model can be extracted from a certain type of data, and what needs to be estimated separately. In addition, our aim is to stress how the choice of the model (for instance choosing a Markovian or a non-Markovian model) affects the predictions one can make based on the data.

We have exposed a method to fit a very general epidemic model to hospital data, using as many analytical tools as we could. As a result, the procedure outlined in this paper, summed up in appendix B, allows one to estimate and monitor the state of an ongoing epidemic all along its course. Indeed, the computing power and time required to fit the model, even with many changes in the contact rate, remains very limited. The main reason for this is that we do not need to explore the space of parameters and simulate the SEIR model for many sets of parameters. Instead, we compute each parameter step by step, using only the information in the exponential fits displayed in figure 3.

As exposed in remark 2.4, knowing the distribution of the sojourn times in each compartment $(\mathcal{E}, \mathcal{I})$ is crucial to deduce the value of $R_0$ from the initial growth rate of the epidemic. Without knowledge on these sojourn times, the reproduction number can only be estimated using contact tracing, which is both difficult and costly. As shown in figure 2, Markovian epidemic models (but not only) can in some regime underestimate the value of $R_0$ for a given initial growth rate of the epidemic. If this is the case, then they will underestimate the final proportion of infected individuals in the population in the absence of containment measures, and thus the predicted death toll of the epidemic.

On the other hand, one needs additional information on the infection fatality ratio (i.e. the proportion of infected individuals who eventually die) and on the distribution of the delay between infection and death in order to estimate the true state of the epidemic at any given time and thus the level of acquired immunity in the population. We have shown that some information on this delay distribution can be retrieved from the time series of hospital deaths caused by the disease, but this information is partly contingent on the model used to estimate the state of the epidemic (that is, on the distribution of $(\mathcal{E}, \mathcal{I})$).

More crucially, one has to wait for some sort of containment measures to take effect in order to estimate these delays, hence this cannot be done during the initial growth phase of the epidemic. Since Markovian models tend to underestimate the inertia of the epidemic after containment measures are put into place (i.e. they tend to decrease exponentially too soon after a change in the contact rate), using these will lead to an underestimation of the length and height of the epidemic *plateau* reached before the decreasing phase. This can be seen in the slow decrease in hospital admissions and deaths even after three weeks of strict lockdown, something which the Markovian ODE model would not have predicted, but that the non-Markovian SEIR model reproduces naturally.

This lack of inertia in ODE models results from the lack of memory property of the exponential distribution: the amount of time an individual already in the E or I compartment stays in this compartment does not depend on the time they have already spent in this compartment, contrary to other more realistic models such as the one considered in the present work. Of course it remains possible to choose the joint distribution of $(\mathcal{E}, \mathcal{I})$ such that the model with memory decreases more rapidly than the ODE model after the start of lockdown measures (for example by choosing Gamma distributions with $k < 1$ for $\mathcal{E}$ and $\mathcal{I}$, see appendix C), but such distributions seem even less realistic than the exponential distribution for infectious disease modelling.

As a result, we expect that using ODE models to fit COVID-19 hospital data forces one to compensate for this in some way. One way to do this would be to make the change in the contact rate more gradual, but we do not believe that this would reflect what took place in France. Alternatively, one could artificially increase the mean exposed time in the model, which results in a longer delay between the change in the contact rate and the change of slope in the daily new infections. However, this would significantly affect the estimates of $R_0$, as can be seen from (2.4). Another way to force the ODE model to fit this behaviour would be to increase the delay between infection and hospital death, but this would also lead to an overestimation of $t_L$, and of the acquired population immunity.

One thing that we can note is that the kind of data studied in this article carries very little information on the actual distribution of $(\mathcal{E}, \mathcal{I})$, at least during the early stages of an epidemic. External information

on the true delay distribution between infection and death, or infection and hospital admission, etc. can help to put constraints on the model, yielding more accurate predictions of the state of the epidemic.

The most obvious simplification in our analysis is the fact that we neglect the age structure in our population. This structure particularly affects the dynamics of COVID-19 since the infection fatality ratio varies significantly with age [23]. Here, we have only used the global infection fatality ratio computed in [23], but a more complex model could be fitted to the age-stratified data provided by Santé Publique France. Moreover, the distribution of the exposed and infectious periods is likely to depend on the age of the infected individual, something which our model fails to consider. Our motivation in neglecting age structure was to keep the model mathematically tractable and to avoid ending up with too many parameters. In order to fully exploit the age-stratified data, one would further need external information on the matrix of contact rates between age-classes, something which is likely to have changed in complex ways as a result of all the containment policies aimed at limiting the spread of the virus.

Note that an analysis similar to the one conducted in this paper can be carried out for a model where the infectivity of each individual is a random function of the time since infection, those random functions for the various individuals being i.i.d., see [20]. This would yield a model which could account for the variability, both in time and between individuals, of the infectivity of each infected individual, something which is apparent for COVID-19 for example in [17]. Applying this type of models to the ongoing COVID-19 epidemic, and including age-structure, will be the subject of future work.

Data accessibility. The code used to simulate the SEIR model and to analyse the data is made available at https://github.com/rforien/Fit_COVID19_nonMarkovian.git. The public sources where the data can be found are indicated in the paper.

Authors' contributions. G.P. and E.P. are the authors of the original model which is used in this paper. Proposition 2.2 was proved by R.F., who also collected the data, made the computer simulations, and wrote the paper. R.F., G.P. and E.P. discussed and corrected the first version, which led to the present version.

Competing interests. We declare we have no competing interests.

Funding. No funding has been received for this article.

# Appendix A. Proof of proposition 2.2

We start with the case $\rho > 0$. Replacing $(E(t), I(t), R(t))$ in (2.2) by their expressions in (2.3) yields the following equations

$$e\,e^{\rho t} = \lambda i \int_{-\infty}^{t} G^c(t-s)\,e^{\rho s}\,\mathrm{d}s$$

$$i\,e^{\rho t} = \lambda i \int_{-\infty}^{t} \Psi(t-s)\,e^{\rho s}\,\mathrm{d}s$$

and

$$r\,e^{\rho t} = \lambda i \int_{-\infty}^{t} \Phi(t-s)\,e^{\rho s}\,\mathrm{d}s.$$

The second equation translates into

$$\begin{aligned}
\lambda^{-1} &= \int_0^{\infty} \mathbb{P}(\mathcal{E} \leq s < \mathcal{E} + \mathcal{I})\,e^{-\rho s}\,\mathrm{d}s \\
&= \mathbb{E}\left[\int_{\mathcal{E}}^{\mathcal{E}+\mathcal{I}} e^{-\rho s}\,\mathrm{d}s\right] \\
&= \frac{\mathbb{E}[e^{-\rho \mathcal{E}}(1 - e^{-\rho \mathcal{I}})]}{\rho},
\end{aligned} \tag{A 1}$$

yielding (2.4). The remaining two equations give the relations between $e$, $r$ and $i$. The first one yields

$$\begin{aligned}
e &= \lambda i \int_0^{\infty} \mathbb{P}(\mathcal{E} > s)\,e^{-\rho s}\,\mathrm{d}s \\
&= \frac{\lambda i}{\rho}(1 - \mathbb{E}[e^{-\rho \mathcal{E}}]),
\end{aligned} \tag{A 2}$$

and the third one gives, for $\rho > 0$,

$$r = \lambda i \int_0^\infty \mathbb{P}(\mathcal{E} + \mathcal{I} \leq s)\, e^{-\rho s}\, ds$$

$$= \frac{\lambda i}{\rho} \mathbb{E}[e^{-\rho(\mathcal{E} + \mathcal{I})}].$$

Combining these with the constraint $e + i + r = 1$ yields (2.5). Note that $\lambda i = \rho$.

To prove the second part of the statement, we only need to check that the quantities defined in (2.6) satisfy

$$e G_0^c(t) = \lambda i \int_{-\infty}^0 G^c(t-s) e^{\rho s}\, ds$$

$$i F_1^c(t) + e \Psi_0(t) = \lambda i \int_{-\infty}^0 \Psi(t-s) e^{\rho s}\, ds$$

$$r + i F_1(t) + e \Phi_0(t) = \lambda i \int_{-\infty}^0 \Phi(t-s) e^{\rho s}\, ds.$$

Indeed, combining these with (2.2) and the first part of the proposition, we obtain the fact that (2.3) solves (2.1). To check the first identity, write

$$\int_{-\infty}^0 G^c(t-s) e^{\rho s}\, ds = \int_t^{+\infty} \mathbb{P}(\mathcal{E} > s)\, e^{\rho(t-s)}\, ds$$

$$= \mathbb{E}\left[ \mathbf{1}_{\{\mathcal{E} > t\}} \int_t^{\mathcal{E}} e^{\rho(t-s)}\, ds \right]$$

$$= \frac{1}{\rho} \mathbb{E}[\mathbf{1}_{\{\mathcal{E} > t\}}(1 - e^{-\rho(\mathcal{E}-t)})].$$

Multiplying by $\lambda i / e$, we obtain

$$G_0^c(t) = \frac{\lambda i}{e} \int_{-\infty}^0 G^c(t-s) e^{\rho s}\, ds = \frac{\mathbb{E}[\mathbf{1}_{\{\mathcal{E} > t\}}(1 - e^{-\rho(\mathcal{E}-t)})]}{\mathbb{E}[1 - e^{-\rho \mathcal{E}}]}.$$

We conclude by noting that the term on right-hand side equals $\mathbb{P}(\mathcal{E} - \Theta > t \mid \Theta < \mathcal{E})$, where $\Theta$ is an independent exponential random variable with parameter $\rho$.

Plugging (2.6) and (2.7) in the second equation, we obtain

$$i F_1^c(t) + e \Psi_0(t) = \mathbb{P}(\mathcal{E} \leq \Theta < \mathcal{E} + \mathcal{I})\, \mathbb{P}(t + \Theta < \mathcal{E} + \mathcal{I} \mid \mathcal{E} \leq \Theta < \mathcal{E} + \mathcal{I})$$
$$+ \mathbb{P}(\theta < \mathcal{E})\, \mathbb{P}(\mathcal{E} \leq t + \Theta < \mathcal{E} + \mathcal{I} \mid \Theta < \mathcal{E}).$$

Since, for $t \geq 0$,

$$\{t + \Theta < \mathcal{E} + \mathcal{I}\} \cap \{\mathcal{E} \leq \Theta < \mathcal{E} + \mathcal{I}\} = \{\mathcal{E} \leq t + \Theta < \mathcal{E} + \mathcal{I}\} \cap \{\mathcal{E} \leq \Theta < \mathcal{E} + \mathcal{I}\}$$

and

$$\{\mathcal{E} \leq t + \Theta < \mathcal{E} + \mathcal{I}\} \cap \{\mathcal{E} + \mathcal{I} \leq \Theta\} = \emptyset,$$

by the law of total probability,

$$i F_1^c(t) + e \Psi_0(t) = \mathbb{P}(\mathcal{E} \leq t + \Theta < \mathcal{E} + \mathcal{I}).$$

On the other hand, since $\lambda i = \rho$,

$$\lambda i \int_{-\infty}^0 \Psi(t-s) e^{\rho s}\, ds = \int_0^{+\infty} \mathbb{P}(\mathcal{E} \leq t + s < \mathcal{E} + \mathcal{I}) \rho\, e^{-\rho s}\, ds$$

$$= \mathbb{P}(\mathcal{E} \leq t + \Theta < \mathcal{E} + \mathcal{I}).$$

For the third equation, we proceed in the same way, plugging (2.6) and (2.7), we obtain

$$r + i F_1(t) + e \Phi_0(t) = \mathbb{P}(\mathcal{E} + \mathcal{I} \leq \Theta) + \mathbb{P}(\mathcal{E} \leq \Theta < \mathcal{E} + \mathcal{I})\, \mathbb{P}(\mathcal{E} + \mathcal{I} \leq t + \Theta \mid \mathcal{E} \leq \Theta < \mathcal{E} + \mathcal{I})$$
$$+ \mathbb{P}(\Theta < \mathcal{E})\, \mathbb{P}(\mathcal{E} + \mathcal{I} \leq t + \Theta \mid \Theta < \mathcal{E}).$$

Now, we note that, for $t \geq 0$,

$$\{\mathcal{E} + \mathcal{I} \leq t + \Theta\} \cap \{\mathcal{E} + \mathcal{I} \leq \Theta\} = \{\mathcal{E} + \mathcal{I} \leq \Theta\}.$$

As a result, applying the law of total probability, we obtain

$$\boldsymbol{r} + \boldsymbol{i}F_1(t) + \boldsymbol{e}\Phi_0(t) = \mathbb{P}(\mathcal{E} + \mathcal{I} \leq t + \Theta).$$

On the other hand, since $\lambda\boldsymbol{i} = \rho$,

$$\lambda\boldsymbol{i} \int_{-\infty}^{0} \Phi(t - s)\,\mathrm{e}^{\rho s}\,\mathrm{d}s = \int_{0}^{+\infty} \mathbb{P}(\mathcal{E} + \mathcal{I} \leq t + s)\rho\,\mathrm{e}^{\rho s}\,\mathrm{d}s$$

$$= \mathbb{P}(\mathcal{E} + \mathcal{I} \leq t + \Theta).$$

This concludes the proof of the first part of proposition 2.2.

In the case $\rho < 0$, note that the computations in (A 1) and (A 2) remain valid, yielding the second part of the statement.

# Appendix B. Estimation procedure

Let us sum up the main steps of the estimation procedure proposed in the present work.

(i) The first step is to define the set of dates $d_1, d_2, \ldots, d_k$ at which the contact rate $\lambda(t)$ changes in the population. This is done using external information (e.g. dates at which lockdowns come into force, dates at which preventive measures change, etc.). These dates correspond to times $t_1, t_2, \ldots, t_k$, where $t_i$ is the time (in days) elapsed since the start of the epidemic at $d_i$ (here $t_1 = t_L$, the time elapsed at the start of the first lockdown). Since the time of the start of the epidemic is yet unknown, at this point one only knows the value of $t_{i+1} - t_i$ for $i \geq 1$.

(ii) We then estimate the growth rate of the epidemic $\rho_i$ between each successive pair of dates of change of $\lambda(t)$, by fitting a set of exponential curves to the cumulative hospital data in each phase. Hence, $\rho_0$ is the growth rate before $d_1$, $\rho_1$ is the growth rate between $d_1$ and $d_2$, and so on. The results of this step are depicted in figure 3. This step also yields an estimate of the cumulative number of deaths at $t_1 = t_L$, $\Lambda_D(t_L)$, and the two constants $A$ and $B$ which appear in (2.16).

(iii) Using the growth rates $\rho_i$, $i \geq 0$, one then computes the corresponding effective contact rates $\lambda_{e,i}$, $i \geq 0$ using proposition 2.2. Since we have assumed that $\overline{S}(t) \approx 1$ during the first two phases (i.e. up to $t_2$), we have $\lambda_0 = \lambda_{e,0}$ and $\lambda_1 = \lambda_{e,1}$.

(iv) The next step is to estimate the distribution of the delays between infection and hospital admission, ICU admission and hospital death, following the procedure of §2.7. The quantities $\rho_0$, $\rho_1$, $A$, $B$ and $\Lambda_D(t_L)$ have been obtained in step 2 and $\lambda_1$ has been obtained in step 3. On the other hand, the constants $C_E$, $C_I$, $C_R$, defined in (2.10), depend only on $\delta$, $\rho_0$, $\rho_1$ and the distribution of $(\mathcal{E}, \mathcal{I})$, and can be computed easily by numerically solving the system of definition 2.1. Using this, one can numerically invert (2.16) to find suitable parameters $\theta$, $k$ for each delay distribution. The result of this step is displayed in figure 4, see also table 4.

(v) After this, one can also compute the hospital admission to infection ratio $p_H$ defined in §2.8, using (2.17), and the corresponding ratio for ICU admission, both from the infection fatality ratio $f$ which needs to be obtained from some external source.

(vi) The only unknown left at this point is the date of the start of the epidemic, $d_0$, which is obtained from (2.15) and the estimated parameters of the delay distribution between infection and hospital death (recall that $t_L$ is the time elapsed since the start of the epidemic at the start of lockdown measures, i.e. between $d_0$ and $d_1$).

(vii) Finally, we solve the system of definition 2.1 where the function $\lambda(t)$ is obtained in the following recursive way. We first set $\lambda(t) = \lambda_0$ for $t \leq t_L$, and solve the system up to $t_L = t_1$. After that, suppose that we have solved the system up to time $t_i$, corresponding to the date $d_i$, for some $i \geq 1$. We then let

$$\lambda(t) = \lambda_i := \frac{\lambda_{e,i}}{\overline{S}(t_i)}, \quad t_i \leq t < t_{i+1},$$

and we solve the system up to time $t_{i+1}$.

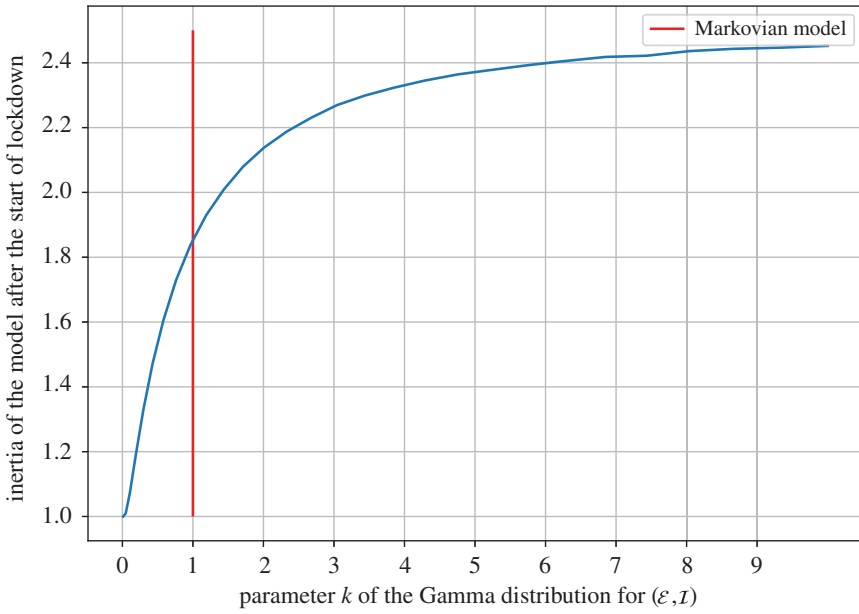

**Figure 7.** Inertia of the model of definition 2.1 around a change in the contact rate, measured by the ratio (C 1). For each value of $k$, we consider that $\mathcal{E}$ and $\mathcal{I}$ are independent Gamma random variables with parameters $(k, m_E/k)$ and $(k, m_I/k)$, respectively, with $m_E = 2$ and $m_I = 5$. The Markovian ODE model corresponds to $k = 1$ (red vertical line). We can see that, for this choice of distributions, decreasing the variance of $(\mathcal{E}, \mathcal{I})$ increases the inertia of the model with memory.

Once the trajectory of the process $(\overline{S}(t), \overline{E}(t), \overline{I}(t), \overline{R}(t))_{t \geq 0}$ has been obtained, we can compute the predicted cumulative and daily number of hospital deaths $(\Lambda_D(t), t \geq 0)$, hospital admissions and ICU admissions in the model, using (2.13). These predictions can then be compared to the original data used in step 2. The result of this step is displayed in figure 5.

## Appendix C. Inertia of the model with memory

One way to measure the inertia of the model is through the ratio

$$\frac{1 - \overline{S}(t_L + \delta)}{1 - \overline{S}(t_L)},\tag{A 3}$$

where $t_L$ is the time of the start of lockdown measures, and $\delta > 0$ is large enough that, at time $t_L + \delta$, the SEIR model is close to the exponential regime associated with the new growth rate, i.e. it satisfies (2.9) (in the present case, $\delta$ is of the order of a few weeks). Indeed, the longer it takes for daily new infections to start decreasing, the higher this ratio will be.

This ratio is displayed in figure 7 for the model of definition 2.1 where

$$(\mathcal{E}, \mathcal{I}) \sim \Gamma\left(k, \frac{m_{\mathcal{E}}}{k}\right) \otimes \Gamma\left(k, \frac{m_{\mathcal{I}}}{k}\right),$$

and $m_E = 2$ days and $m_I = 5$ days, for several values of $k \in (0, 10]$. In this way, $(\mathcal{E}, \mathcal{I})$ is a pair of independent Gamma random variables, with respective expectations $m_E$ and $m_I$, and the parameter $k$ controls the ratio between the variance and the expectation of these random variables. We observe on this figure that the inertia of the model seems to be an increasing function of the parameter $k$, i.e. it increases when the variance of $\mathcal{E}$ and $\mathcal{I}$ decreases. Note that the case $k = 1$ corresponds to $(\mathcal{E}, \mathcal{I})$ being a pair of exponential random variables, i.e. the Markovian ODE model.

While we do not assume that $\mathcal{E}$ and $\mathcal{I}$ follow Gamma distributions in this work, figure 7 illustrates how the distribution of $(\mathcal{E}, \mathcal{I})$ affects the inertia of the model with memory of definition 2.1.

## Appendix D. Sensitivity analysis

We performed a Sobol sensitivity analysis using the SALib python package to investigate the effect of the distribution of $(\mathcal{E}, \mathcal{I})$ on the inferred value of $R_0$. We let the mean exposed period vary between 2 and 5

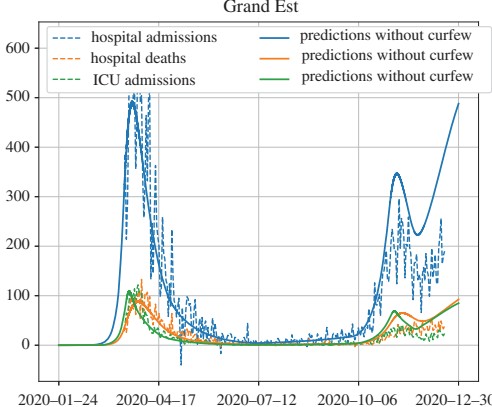
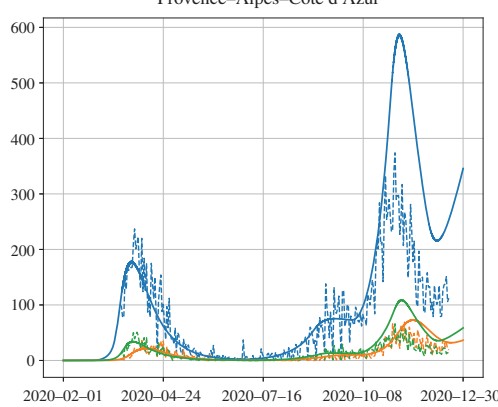

**Figure 8.** Model predictions of hospital admissions, ICU admissions and hospital deaths in the Grand Est and PACA regions without accounting for the (putative) effect of curfew before the second lockdown. To compute this, we assumed a single change of contact rate on 30 October, with the rate computed from the halving time during the second lockdown.

**Table 5.** First-order Sobol indices of $R_0$ relative to the three parameters.

| parameter | first-order Sobol indices |
| --- | --- |
| mean exposed period | $0.655 \pm 0.019$ |
| proportion of reported individuals | $0.174 \pm 0.012$ |
| mean unreported infectious period | $0.165 \pm 0.011$ |

days, the proportion of reported individuals between 0.2 and 0.8, and we let the mean infectious period of unreported individuals vary between 8 and 12 days. The resulting Sobol indices are given in table 5. The parameters turned out to have very little interactions, with upper bounds for the confidence interval lying in the range 0.018–0.03.

Unsurprisingly, the parameter with the largest effect on the value of $R_0$ is the mean exposed period, as can already be seen in corollary 2.3.

# Appendix E. Change of the contact rate before the second lockdown

Figure 8 shows the result of the model predictions in the Grand Est and PACA regions when we assume a single change of the contact rate on 30 October at the start of the second lockdown. As we can see, the model grossly overestimates the peak during the second lockdown. This means that the contact rate must have been reduced slightly even before the start of the lockdown. This coincides with the time at which a nightly curfew was enforced in several large cities, including Marseilles and Strasbourg, the two largest cities in these regions. However, this does not mean that the curfew in itself is necessarily responsible for the reduction of the contact rate.

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
