## [Peer Review File · Royal Society Open Science]

Review History

RSOS-202327.R0 (Original submission)

Review form: Reviewer 1

Is the manuscript scientifically sound in its present form?

Yes

Are the interpretations and conclusions justified by the results?

Yes

Is the language acceptable?

Yes

Do you have any ethical concerns with this paper?

No

Have you any concerns about statistical analyses in this paper?

No

Recommendation?

Accept with minor revision (please list in comments)

Comments to the Author(s)

Please see the attached file (Appendix A).

Review form: Reviewer 2

Is the manuscript scientifically sound in its present form?

Yes

Are the interpretations and conclusions justified by the results?

Yes

Is the language acceptable?

Yes

Do you have any ethical concerns with this paper?

No

Have you any concerns about statistical analyses in this paper?

No

Recommendation?

Accept with minor revision (please list in comments)

Comments to the Author(s)

In this article, the authors introduce a SEIR-type epidemic model to be applied on the evolution of the Covid-19 epidemics in France in 2020. This model is based on an individual-based system in which a susceptible individual is contaminated at Poisson rate λ by contaminated individuals. The individual remains a random time E in an exposed phase, before becoming infected for a random time I , during which it can contaminate further susceptible individuals. After that time, the individual is recovered, and removed from consideration. The SEIR model is then chosen as the large scale limit of this individual-based dynamics.

Contrarily to typical SEIR models, the pair of random time (E, I) is chosen neither independent nor exponentially distributed. As a result, the evolution of the individual between its exposed and its recovered stage is not Markovian. As a result, the macroscopic model is not described by a system of ODE but by a more complicate system of integral equations.

In this article, the authors show in a very pedagogic way how to use the limited amount of information given by the number of hospitalization and deaths to estimate the parameters of the system. They use it in particular to estimate the base reproduction number (R_0) of the epidemic during the different stages of growth and lock-down. It is worth noting that the choice of a non-Markovian system increases the estimated value of the R_0 compared to Markovian systems.

The article is very interesting and its application to the freely available data of the epidemics is very on point. I would therefore recommend acceptance of the present article in {Royal Society Open Science}, provided the following minor points are addressed.

Decision letter (RSOS-202327.R0)

Dear Dr Pardoux

On behalf of the Editors, we are pleased to inform you that your Manuscript RSOS-202327 "Estimating the state of the Covid-19 epidemic in France using a non-Markovian model" has been accepted for publication in Royal Society Open Science subject to minor revision in accordance with the referees' reports. Please find the referees' comments along with any feedback from the Editors below my signature.

Please submit your revised manuscript and required files (see below) no later than 7 days from today's (ie 10-Feb-2021) date. Note: the ScholarOne system will 'lock' if submission of the revision is attempted 7 or more days after the deadline. If you do not think you will be able to meet this deadline please contact the editorial office immediately.

Kind regards,

Anita Kristiansen
Editorial Coordinator

on behalf of Dr Pierre Magal (Associate Editor) and Glenn Webb (Subject Editor)
openscience@royalsociety.org

Reviewer comments to Author:
Reviewer: 1

Comments to the Author(s)
Please see the attached file.

Reviewer: 2

Comments to the Author(s)

In this article, the authors introduce a SEIR-type epidemic model to be applied on the evolution of the Covid-19 epidemics in France in 2020. This model is based on an individual-based system in which a susceptible individual is contaminated at Poisson rate λ by contaminated individuals. The individual remains a random time E in an exposed phase, before becoming infected for a random time I , during which it can contaminate further susceptible individuals. After that time, the individual is recovered, and removed from consideration. The SEIR model is then chosen as the large scale limit of this individual-based dynamics.

Contrarily to typical SEIR models, the pair of random time (E, I) is chosen neither independent nor exponentially distributed. As a result, the evolution of the individual between its exposed and its recovered stage is not Markovian. As a result, the macroscopic model is not described by a system of ODE but by a more complicate system of integral equations.

In this article, the authors show in a very pedagogic way how to use the limited amount of information given by the number of hospitalization and deaths to estimate the parameters of the system. They use it in particular to estimate the base reproduction number (R_0) of the epidemic during the different stages of growth and lock-down. It is worth noting that the choice of a non-Markovian system increases the estimated value of the R_0 compared to Markovian systems.

The article is very interesting and its application to the freely available data of the epidemics is very on point. I would therefore recommend acceptance of the present article in {Royal Society Open Science}, provided the following minor points are addressed.

===PREPARING YOUR MANUSCRIPT===

If you have been asked to revise the written English in your submission as a condition of publication, you must do so, and you are expected to provide evidence that you have received

language editing support. The journal would prefer that you use a professional language editing service and provide a certificate of editing, but a signed letter from a colleague who is a native speaker of English is acceptable. Note the journal has arranged a number of discounts for authors using professional language editing services (<https://royalsociety.org/journals/authors/benefits/language-editing/>).

===PREPARING YOUR REVISION IN SCHOLARONE===

Author's Response to Decision Letter for (RSOS-202327.R0)

See Appendices B - D.

Decision letter (RSOS-202327.R1)

Dear Dr Pardoux,

It is a pleasure to accept your manuscript entitled "Estimating the state of the Covid-19 epidemic in France using a model with memory" in its current form for publication in Royal Society Open Science.

COVID-19 rapid publication process:

We are taking steps to expedite the publication of research relevant to the pandemic. If you wish, you can opt to have your paper published as soon as it is ready, rather than waiting for it to be published the scheduled Wednesday.

This means your paper will not be included in the weekly media round-up which the Society sends to journalists ahead of publication. However, it will still appear in the COVID-19 Publishing Collection which journalists will be directed to each week (<https://royalsocietypublishing.org/topic/special-collections/novel-coronavirus-outbreak>).

If you wish to have your paper considered for immediate publication, or to discuss further, please notify openscience_proofs@royalsociety.org and press@royalsociety.org when you respond to this email.

on behalf of Dr Pierre Magal (Associate Editor) and Glenn Webb (Subject Editor)
openscience@royalsociety.org

Appendix A

Reviewer comments on manuscript “Estimating the state of the COVID-19 epidemic in France using a non-Markovian model”

In this paper, the authors derived methods for estimating a few important epidemiological parameters from a non-Markovian SEIRU (U for unidentified cases) model. The structure and aim of this paper is similar to the one about parameter estimations of the SEIRU model with constant coefficients by [Liu, Magal, Seydi, Webb] (and other related papers from the same group), and in this paper the authors obtained results for a non-Markovian version. The key idea of their method is that the exponential growth rate, ρ , of an epidemic can be firstly obtained by a linear regression fitting to the outbreak data. Then all the other epidemiological parameters - such as stage-specific contact rates, shifting time points of intervention stages, and mean values related to the incubation period, infectiousness period, hospitalization rates, and death rates - can be directly computed from the formulas derived from this study.

The authors firstly applied numerical simulations to compare the estimation of R_0 from different models with various assumptions on the incubation and infectiousness distribution functions, and this comparison was implemented under fixed epidemic growth rate ρ . The authors concluded from this experiment that the traditional SEIR models with constant coefficients could underestimate R_0 under such scenario. In the Results section, the authors applied their method and utilized outbreak data in France to provide their estimations of the epidemiological parameters.

This paper is clearly presented, and the derivations of all formulas are straight forward to readers. The fitting methods should be applicable for inferring the epidemiological characteristics of an emerging disease outbreak. Here are some suggestions for the authors to consider in their revised version.

1. Concerns on estimating the cumulative number of deaths: the authors assumed in equation (11) that the cumulative deaths is proportional to the number of all infected individuals, which is an odd assumption in two aspects. Firstly, it is not normal for people to die outside of hospitals, although such events are still possible, the fraction of people dying outside of the hospitals (i.e. those people in E and I classes) should be far less than that of those admitted to hospitals (i.e. those in R class). Secondly, if the authors assume people in the E and I classes are dying, then the deceased individuals would not contribute to the transmission dynamics, thus the dynamics of dying should be included in the original model as extra terms.
2. The first paragraph on page 4 for introducing the random variables is confusing: the authors could consider listing definitions for all random variables involved (E, I, E0, I0, I1), and make the definitions more concise.
3. The major goal of this paper is providing a strategic method to estimate the parameters. The authors could consider summarizing their methods as an algorithm to facilitate the readers in applying their method.

Appendix B

Estimating the state of the Covid-19 epidemic in France using a ~~non-Markovian~~ model with memory

Raphaël Forien^a, Guodong Pang^b, and Étienne Pardoux^c

^aINRAE, BioSP, Centre INRAE PACA, Domaine St-Paul, 84914 Avignon Cedex FRANCE, raphael.forien@inrae.fr

^bThe Harold and Inge Marcus Department of Industrial and Manufacturing Engineering, College of Engineering, Pennsylvania State University, University Park, PA 16802 USA, gup3@psu.edu

^cAix-Marseille Université, CNRS, Centrale Marseille, I2M, UMR 7373 13453 Marseille, France, etienne.pardoux@univ.amu.fr

February 16, 2021

Abstract

In this paper, we use a deterministic epidemic model with memory to estimate the state of the Covid-19 epidemic in France. ~~This model allows us to consider realistic distributions for the~~, from early March until mid December 2020. Our model is in the SEIR class, which means that when a susceptible individual (S) becomes infected, he/she is first exposed (E), i.e., not yet contagious. Then he/she becomes infectious (I) for a certain length of time, during which he/she may infect susceptible individuals around him/her, and finally becomes removed (R), that is, either immune or dead. The specificity of our model is that it assumes a very general probability distribution for the pair of exposed ~~and infectious periods in a SEIR model, contrary to standard ODE models which only consider exponentially distributed exposed~~ and infectious periods. The law of large numbers limit of such a model is a model with memory (the future evolution of the model depends not only upon its present state, but also upon its past). We present theoretical results linking the (unobserved) parameters of the model to various quantities which are more easily measured during the early stages of an epidemic. We ~~also stress the main quantitative differences between the deterministic models which are large-population limits of non-Markovian and Markovian models.~~ We then apply these results to estimate the state of the Covid-19 epidemic in France ~~by analyzing the evolution of the epidemic separately in three distinct regions: the Paris region, the northeast region and the rest of the country, based on current knowledge on~~, using available information on the infection fatality ratio and ~~the~~ on the distribution of the exposed and infectious periods ~~distributions for Covid-19. Our analysis is based on~~. Using the hospital data published daily by Santé Publique France ~~(daily hospital admissions, intensive care unit admissions~~, we gather some information on the delay between infection and hospital admission, ICU admission and hospital deaths), and on the proportion of people who have been infected up to the end of 2020.

Keywords: Covid-19, model with memory, Volterra equations.

Introduction

At the beginning of an epidemic outbreak, some quantities are easier to observe and report than others. For example, the number of hospital deaths related to Covid-19 has been precisely and regularly reported in several countries. Another well documented quantity is the doubling time of the number of cases (which coincides with that of the number of deaths, as we shall explain). On the other hand, some quantities of interest are very hard to directly measure or to estimate: the actual number of infected individuals, the true death rate of the disease, and most notably the now famous **basic** reproduction number R_0 .

This paper presents a theoretical study of how the “observable” quantities relate to the “unobserved” ones, under a very general epidemic model recently developed in [1], and an application of these results to attempt to describe and predict the evolution and current state of the Covid-19 epidemic in France.

The first basic idea is to estimate an exponential growth rate from the available data (the number of hospital admissions or the number of deaths occurring in hospitals), and then to relate it to the parameters of a model of the evolution of the Covid-19 epidemic. Our approach is modeled upon the approach of Tom Britton in two recent preprints [2], [3]. Note that the exponential growth rate can be negative, which was the case in particular in France during the two lockdown periods.

The exponential growth rate is the most natural parameter which can be extracted from the data. It equals $\log(2)$ divided by the doubling time, i.e., the number of days necessary for the number of daily new cases (or deaths) to double, a notion of which everyone listening to the news at the time of the rise of the epidemic has heard. The other important parameters of the model, including the famous basic reproduction number R_0 (the mean number of individuals whom an infectious individual infects before recovering, at the start of the epidemic – that is, while essentially everyone ~~that~~ around them is susceptible), can be computed from the exponential growth rate, and some other parameters of the model.

Our approach has two specific features. First, we shall carefully make explicit which parameters are needed to compute the quantities of interest, and what information can be retrieved from the available data if these parameters are known. The second aspect is that we shall use nonconventional models. The classical SIR and SEIR ODE models are law of large numbers limits of stochastic Markov SIR or SEIR models [4]. For those models to be Markovian, it is necessary that the infectious periods in the case of the SIR model of the various individuals in the population be i.i.d. copies of an exponential random variable (resp. the pair exposed period, infectious period in the SEIR model be i.i.d. copies of a pair of independent exponential random variables). However, in the case of Covid-19, like in most infectious diseases, the assumption that those durations follow an exponential distribution is unrealistic.

Recently, the last two authors of this paper have described the law of large numbers limit model of non-Markovian stochastic epidemic models with arbitrary distribution for the infectious period (resp. for the pair exposed and infectious period), see [1]. This type of deterministic model is an integral equation of Volterra type, of the same dimension as the classical ODE model. In particular, it is not much more complicated to simulate and compute. The differences between our model and the usual “Markov” model are not so much in the large time behavior, but rather in the transient short term evolution, as was recently

observed in [5], who study a discrete time model with memory effects in order to account for the inertia of the epidemic and the delays before hospital admission and death. We note that since the start of the epidemic, knowledge about the Covid-19 disease has substantially increased. Our original approach allows us to choose distributions which reflect closely the current knowledge about the disease.

Note that a handful of other works have used similar extended models to analyze the Covid-19 epidemic. In [6], the authors use ODEs with delays, which correspond to our model with deterministic exposed and infectious periods. On the other hand, following the approach initiated by Kermack and McKendrick to analyze the plague epidemic in Mumbai in 1905–6 [7], [8] uses a transport PDE SEIR model, where the rate of infection by an infectious individual depends upon the time since infection, and the rate at which exposed (resp. infectious) individuals become infectious (resp. removed) also depends on the time since infection. The results of the present paper are comparable to those in [8], where the analysis and the treated data concern the various *départements* of Île de France, while we compare Île de France, Grand Est, Provence–Alpes–Côte d’Azur and Auvergne–Rhône–Alpes. Finally, [9] ~~use~~-uses an age-of-infection structure to model the Covid-19 epidemic in France.

Models similar to our non-Markovian SEIR epidemic model appear in several papers (see, e.g., [10, 11]) and in section 4.5 of the recent book [12]. See also [13], which compares a SIR model with delay (which corresponds at the level of the individual based model to a deterministic infectious period) with the classical SIR ODE model. However, our model seems to be more general than those ~~who~~-which appeared earlier, in particular regarding the initial condition and allowing correlated exposed and infectious periods, and as far as we can tell its rigorous interpretation as the functional law of large numbers limit of an individual based stochastic model established in [1] is new. This interpretation was crucial to suggest the reduction of the dimension of the model which we shall describe below. As a matter of fact, our formula for R_0 may be considered as a particular case of a formula which appears on page 141 of [12] (see also [14] and [15]).

The paper is organized as follows. In a first section called “Methods”, we describe our model and the methodology to extract the parameters of our model from the available data. In a second section called “Results”, we describe the results we obtained by applying our method to the Covid-19 epidemic in France. The last section is a discussion of the conclusions of our work. Finally, an appendix contains the mathematical proof of one crucial result, which relates the exponential growth rate to the various quantities in our model.

1 Methods

1.1 ~~A non-Markovian~~-An SEIR epidemic model with memory for Covid-19

We shall use a deterministic SEIR epidemic model with memory. That model is shown in [1] to be the law of large numbers limit, as the population size tends to infinity, of an individual-based stochastic model, which we first describe.

Assume that each newly infected individual in the population becomes infectious after a

random time \mathcal{E} during which ~~they are he/she~~ is “exposed” and ~~stops being infectious after~~ stays infectious for a random time \mathcal{I} , after which ~~they do he/she~~ does not infect anyone any more, and also cannot be infected (either as a result of acquired immunity, isolation or death). To each individual is associated a copy $(\mathcal{E}_i, \mathcal{I}_i)$ of the random pair $(\mathcal{E}, \mathcal{I})$, in such a way that the $(\mathcal{E}_i, \mathcal{I}_i)_{i \geq 1}$ are independent and identically distributed (abbreviated i.i.d.). The important point which distinguishes our model from most common epidemic models is that we do not assume that the two random variables \mathcal{E} and \mathcal{I} are independent nor that they follow exponential distributions. Hence the stochastic model is not a Markov model, but rather a model with memory, which will also be the case of the limiting model. We assume that infectious individuals attempt to infect other individuals (chosen uniformly from the population) at rate $\lambda(t)$, where t is the current time (the dependence on t of the contact rate λ is used to reflect the effect of containment measures such as lockdown, and social behaviour such as use of face masks and physical distancing). As a result, the ~~process~~ number of newly infected individuals up to time t can be represented as

$$P \left(N \int_0^t \lambda(s) \bar{S}^N(s) \bar{I}^N(s) ds \right),$$

where $P(t)$ is a standard Poisson process, N is the size of the population which is assumed to be fixed ~~throughout the epidemic (the deaths due to the epidemic are counted among the “removed”)~~, $\bar{S}^N(t)$ (resp. $\bar{I}^N(t)$) denotes the proportion of susceptible (resp. infectious) individuals in the population at time t .

~~Assume that the initial proportions of susceptible, exposed, infectious~~ In addition, let $(\bar{E}^N(t), \bar{R}^N(t))$ denote the proportions of exposed and removed individuals at time t in the population ~~are given by $(\bar{S}(0), \bar{E}(0), \bar{I}(0), \bar{R}(0))$. We also assume that the individuals who are~~, respectively.

The initially exposed or infectious ~~have a different distribution of sojourn times in the intermediary states (reflecting the fact that they have been infected at some time~~ individuals are thought of as having been infected in the past ~~, before the initial time $t = 0$). Thus let $(\mathcal{E}_0, \mathcal{I}_0)$ denote a random variable distributed as the time an individual who is initially exposed stays exposed (\mathcal{E}_0) and then infectious (\mathcal{I}_0), and let \mathcal{I}_1 be distributed as the time an individual who is initially infectious stays infectious. Note that we may ~~and do in the following~~ assume that \mathcal{I}_0 . While in the case of exponential exposed and \mathcal{I} -infectious periods, it is quite natural to assume that the remaining exposed (resp. infectious) periods of the initially exposed (resp. infectious) individuals have the same ~~distribution for those who are initially exposed, while the remaining infectious period \mathcal{I}_1 may have a different distribution. Similarly~~ law as those of the individuals infected at time $t > 0$ (due to the lack of memory property of the exponential distribution), that is no longer the case in our model.~~

Therefore we need to introduce the distribution of the pair $(\mathcal{E}_0, \mathcal{I}_0)$, the remaining exposed period \mathcal{E}_0 may have a different distribution from that of \mathcal{E} , and we assume that \mathcal{E}_0 and infectious period of an initially exposed individual, as well as the distribution \mathcal{I}_1 of the remaining infectious period of an initially infectious individual. More precisely, if $E(0)$ (resp. $I(0)$) denotes the number of initially exposed (resp. infectious) individuals, then the collection of random variables $(\mathcal{E}_{0,i}, \mathcal{I}_{0,i})_{1 \leq i \leq E(0)}$, (resp. $(\mathcal{I}_{1,i})_{1 \leq i \leq I(0)}$) is supposed to be i.i.d.

and describes the duration of the remaining exposed and infectious periods of the initially exposed individuals (resp. the duration of the remaining infectious periods of the initially infectious individuals). We of course assume that the three collections $(\mathcal{E}_{0,i}, \mathcal{I}_{0,i})_{1 \leq i \leq E(0)}$, $(\mathcal{I}_{1,i})_{1 \leq i \leq I(0)}$ and \mathcal{I}_0 are independent. ~~—~~ $(\mathcal{E}_i, \mathcal{I}_i)_{i \geq 1}$ are mutually independent, and are globally independent of the above Poisson process $P(t)$.

We further let $S(0)$ (resp. $R(0)$) denote the number of initially susceptible (resp. removed) individuals, and define $N = S(0) + E(0) + I(0) + R(0)$, the total population size. For $X = S(0), E(0), I(0)$ or $R(0)$, let $\bar{X} = N^{-1}X$. We assume that as $N \rightarrow \infty$,

$$(\bar{S}^N(0), \bar{E}^N(0), \bar{I}^N(0), \bar{R}^N(0)) \rightarrow (\bar{S}(0), \bar{E}(0), \bar{I}(0), \bar{R}(0))$$

in probability. Define

$$G^c(t) = \mathbb{P}(t < \mathcal{E}), \quad \Psi(t) = \mathbb{P}(\mathcal{E} \leq t < \mathcal{E} + \mathcal{I}), \quad \Phi(t) = \mathbb{P}(\mathcal{E} + \mathcal{I} \leq t),$$

~~as well as~~ as well as

$$G_0^c(t) = \mathbb{P}(t < \mathcal{E}_0), \quad \Psi_0(t) = \mathbb{P}(\mathcal{E}_0 \leq t < \mathcal{E}_0 + \mathcal{I}_0), \quad \Phi_0(t) = \mathbb{P}(\mathcal{E}_0 + \mathcal{I}_0 \leq t),$$

and

$$F_1(t) = \mathbb{P}(\mathcal{I}_1 \leq t), \quad F_1^c(t) = 1 - F_1(t) = \mathbb{P}(t < \mathcal{I}_1).$$

~~Let $(\bar{S}^N(t), \bar{E}^N(t), \bar{I}^N(t), \bar{R}^N(t))$ denote the proportion of susceptible, exposed, infectious and removed individuals at time t in the population, where N is the total size of the population, which is assumed to be fixed throughout the epidemic (the deaths due to the epidemic are counted among the “removed”). We assume that the durations $(\mathcal{E}_i, \mathcal{I}_i)$ associated to the various individuals infected during the epidemic are i. i. d.—~~

Definition 1. Let $\lambda : \mathbb{R}_+ \rightarrow \mathbb{R}_+$ be a bounded càdlàg function. The deterministic non-Markovian SEIR model is the solution of the set of integral equations:

$$\begin{aligned} \bar{S}(t) &= \bar{S}(0) - \int_0^t \lambda(s) \bar{S}(s) \bar{I}(s) ds, \\ \bar{E}(t) &= \bar{E}(0) G_0^c(t) + \int_0^t \lambda(s) G^c(t-s) \bar{S}(s) \bar{I}(s) ds, \\ \bar{I}(t) &= \bar{I}(0) F_1^c(t) + \bar{E}(0) \Psi_0(t) + \int_0^t \lambda(s) \Psi(t-s) \bar{S}(s) \bar{I}(s) ds, \\ \bar{R}(t) &= \bar{I}(0) F_1(t) + \bar{E}(0) \Phi_0(t) + \int_0^t \lambda(s) \Phi(t-s) \bar{S}(s) \bar{I}(s) ds. \end{aligned}$$

Theorem 3.1 from [1] states that, under a very weak assumption on the joint distribution of $(\mathcal{E}, \mathcal{I})$, the unique solution of the above system of integral equations with memory is the functional law of large numbers limit of the above described non-Markovian model. More precisely, as $N \rightarrow \infty$, $(\bar{S}^N(t), \bar{E}^N(t), \bar{I}^N(t), \bar{R}^N(t)) \rightarrow (\bar{S}(t), \bar{E}(t), \bar{I}(t), \bar{R}(t))$ in probability, locally uniformly in t . We call this model non-Markovian since the stochastic process

$(\bar{S}^N(t), \bar{E}^N(t), \bar{I}^N(t), \bar{R}^N(t))$ is not a Markov process in general, because its future evolution depends not only upon its present value, but also upon how long the exposed and infectious individuals have already been in the corresponding **compartment** **compartments**. For the model to be Markovian, it is necessary that the random variables \mathcal{E} and \mathcal{I} be independent and have exponential distributions. In that case, the integral equation SEIR model reduces to the following ODE model

$$\begin{aligned} \frac{d\bar{S}(t)}{dt} &= -\lambda(t)\bar{S}(t)\bar{I}(t), & \frac{d\bar{E}(t)}{dt} &= \lambda(t)\bar{S}(t)\bar{I}(t) - \nu\bar{E}(t), \\ \frac{d\bar{I}(t)}{dt} &= \nu\bar{E}(t) - \gamma\bar{I}(t), & \frac{d\bar{R}(t)}{dt} &= \gamma\bar{I}(t), \end{aligned}$$

where ν (resp. γ) is the parameter of the exponential law of \mathcal{E} (resp. \mathcal{I}), i.e., $\mathbb{E}(\mathcal{E}) = \nu^{-1}$, $\mathbb{E}(\mathcal{I}) = \gamma^{-1}$. Note that in the Markovian case, the convergence $(\bar{S}^N(t), \bar{E}^N(t), \bar{I}^N(t), \bar{R}^N(t)) \rightarrow (\bar{S}(t), \bar{E}(t), \bar{I}(t), \bar{R}(t))$ as $N \rightarrow \infty$ is a classical result, see, e.g., [4] for a recent account. The above ODE model is the one which is used by almost all papers dealing with epidemic models, in particular the Covid-19 models, with the exception, to our knowledge, of [6], where the case of fixed \mathcal{E} and \mathcal{I} is considered and [8, 5, 9]. It is fair to say that the exponential distributions are very poor models for the laws of \mathcal{E} and \mathcal{I} , and it may seem strange that the above ODE model is so widely used, while it is based on **such** unrealistic assumptions. While the large time behavior of the model (e.g., the endemic equilibrium in the case of a SIS or a SIRS model, see **Remarks 2.8 and 3.5** **Section 4.3** in [1]) **depend** **depends** only on the expectation of the random vector $(\mathcal{E}, \mathcal{I})$, and are the same in a Markov and a non-Markov model with identical expected infectious periods, as we shall see below, the transient behavior of the model depends much more on the details of the law of $(\mathcal{E}, \mathcal{I})$.

Let us now explain the specific model (with two variants) which we will use for the Covid-19 epidemic. In our model, the two random variables \mathcal{E} and \mathcal{I} will be independent. The random variable \mathcal{E} will be distributed on the interval $[2, 4]$. Concerning the random variable \mathcal{I} , we assume a bimodal law with support in $[3, 5] \cup [8, 12]$ (details in Subsection 1.4).

The idea behind that type of law **for** \mathcal{I} is as follows. Our model is close to the SEIRU model of [16], which thanks to the flexibility of our class of models we are able to simplify. Indeed, in that paper the individuals are first exposed (state E), then infectious (state I) but without symptoms, and at the end of the I period, a fraction of the individuals are isolated quickly after the onset of symptoms (state R as “reported”), either because they are admitted to hospital or because they self-isolate at home, while the other individuals are not isolated (state U as “unreported”), either because they are unable to do so or because they have light or no symptoms. While in the R class, the individuals are isolated and they do not infect susceptibles any more. For that reason, we identify the “reported” class with part of the “recovered” class, thus reconciling the two meanings of R. Finally, some of the individuals are infectious **while** **when** they are in the I class only, while others are infectious in both I and U classes. In our model, being in the U class is considered as staying longer in the I class (see Figure 1). This is the motivation for the bimodal distribution of the law of \mathcal{I} .

Figure 1: Flow chart of the SEIRU model of [16] and the equivalent SEIR non-Markovian model. In the latter, the two infectious compartments (I and U) are merged into one compartment, but the sojourn time \mathcal{I} of the infectious compartment has a bimodal distribution corresponding to the two subpopulations of reported and unreported individuals. Note that R (reported) and Rem (removed) have been merged into a unique compartment.

1.2 The ~~non-Markovian~~ model with memory during the early phase of the epidemic

During the early phase of the epidemic, the cumulative number of infected individuals remains small compared to the total size of the population. As a result, $\bar{S}(t) \approx 1$ during this phase. Letting $(E(t), I(t), R(t))$ denote the absolute numbers of exposed, infectious and removed individuals during this phase, *i.e.*,

$$(E(t), I(t), R(t)) = (N\bar{E}(t), N\bar{I}(t), N\bar{R}(t)),$$

and assuming that λ is constant during this phase, the non-Markovian SEIR model reduces to

$$\begin{aligned} E(t) &= E(0)G_0^c(t) + \lambda \int_0^t G^c(t-s)I(s)ds, \\ I(t) &= I(0)F_1^c(t) + E(0)\Psi_0(t) + \lambda \int_0^t \Psi(t-s)I(s)ds, \\ R(t) &= R(0) + I(0)F_1(t) + E(0)\Phi_0(t) + \lambda \int_0^t \Phi(t-s)I(s)ds. \end{aligned} \quad (1)$$

The initial state of an epidemic is seldom known, and the Covid-19 epidemic is no exception. However, it is always the case that, once sufficiently many individuals have been infected, the cumulative number of infected individuals grows exponentially with some rate $\rho > 0$. We thus look for solutions to

$$\begin{aligned} E(t) &= \lambda \int_{-\infty}^t G^c(t-s)I(s)ds, \\ I(t) &= \lambda \int_{-\infty}^t \Psi(t-s)I(s)ds, \\ R(t) &= \lambda \int_{-\infty}^t \Phi(t-s)I(s)ds, \end{aligned} \quad (2)$$

for $t \in \mathbb{R}$ which are of the form

$$E(t) = \mathbf{e} e^{\rho t}, \quad I(t) = \mathbf{i} e^{\rho t}, \quad R(t) = \mathbf{r} e^{\rho t}, \quad (3)$$

with $\mathbf{e} + \mathbf{i} + \mathbf{r} = 1$. These equations started from $-\infty$ can be seen as modeling an epidemic which has been growing for a very long time from an infinitesimal proportion of non-susceptible individuals.

One expects (and the results below will confirm that expectation) that ρ is uniquely determined, given λ and the law of $(\mathcal{E}, \mathcal{I})$. However, λ is not known, while ρ is rather easily estimated from the available data, as we will see below, and we shall conjecture a law for $(\mathcal{E}, \mathcal{I})$, based on the observations of medical doctors (see in particular [17] and also [18]). Hence we are interested in the inverse problem of expressing the unknown parameter λ as a function of ρ and the law of $(\mathcal{E}, \mathcal{I})$, which is the first result of the following Proposition.

Proposition 2. (i) *If $\rho > 0$, the system (2) admits solutions of the form (3) for all $t \in \mathbb{R}$ if*

$$\lambda = \frac{\rho}{\mathbb{E}[e^{-\rho\mathcal{E}}(1 - e^{-\rho\mathcal{I}})]} \quad (4)$$

and with

$$\mathbf{e} = \mathbb{E}[1 - e^{-\rho\mathcal{E}}], \quad \mathbf{i} = \mathbb{E}[e^{-\rho\mathcal{E}}(1 - e^{-\rho\mathcal{I}})], \quad \mathbf{r} = \mathbb{E}[e^{-\rho(\mathcal{E}+\mathcal{I})}]. \quad (5)$$

Moreover, if Θ is an independent exponential variable with parameter ρ , then (3) also solves (1) for all $t \geq 0$ with

$$\begin{aligned} G_0^c(t) &= \mathbb{P}(\mathcal{E} > t + \Theta \mid \Theta < \mathcal{E}), & \Psi_0(t) &= \mathbb{P}(\mathcal{E} \leq t + \Theta < \mathcal{E} + \mathcal{I} \mid \Theta < \mathcal{E}), \\ \Phi_0(t) &= \mathbb{P}(t + \Theta \geq \mathcal{E} + \mathcal{I} \mid \Theta < \mathcal{E}), & F_1(t) &= \mathbb{P}(t + \Theta \geq \mathcal{E} + \mathcal{I} \mid \mathcal{E} \leq \Theta < \mathcal{E} + \mathcal{I}). \end{aligned} \quad (6)$$

(ii) If $\rho < 0$, then $(E(t), I(t)) = (\mathbf{e}e^{\rho t}, \mathbf{i}e^{\rho t})$ solves the first two lines of (2) for all $t \in \mathbb{R}$ if λ and ρ satisfy (4) and if

$$\mathbf{e} = \mathbb{E} [e^{-\rho \mathcal{E}} - 1], \quad \mathbf{i} = \mathbb{E} [e^{-\rho \mathcal{E}}(e^{-\rho \mathcal{I}} - 1)].$$

The fact that a solution of the form $R(t) = \mathbf{r}e^{\rho t}$ only exists for positive ρ should not come as a surprise, since $t \mapsto R(t)$ is non-decreasing. Also note that, in the case $\rho > 0$, we can rewrite (5) with the help of the variable Θ as follows,

$$\mathbf{e} = \mathbb{P}(\Theta < \mathcal{E}), \quad \mathbf{i} = \mathbb{P}(\mathcal{E} \leq \Theta < \mathcal{E} + \mathcal{I}), \quad \mathbf{r} = \mathbb{P}(\Theta \geq \mathcal{E} + \mathcal{I}). \quad (7)$$

The proof of Proposition 2 is postponed to Appendix A.

Corollary 3. *The basic reproduction number R_0 , defined as the mean number of secondary infections caused by a single infectious individual in a fully susceptible population, is linked to the initial growth rate of the cumulative number of infected individuals ρ by the relation*

$$R_0 = \lambda \mathbb{E}[\mathcal{I}] = \frac{\rho \mathbb{E}[\mathcal{I}]}{\mathbb{E}[e^{-\rho \mathcal{E}}(1 - e^{-\rho \mathcal{I}})]}.$$

This formula remains valid if $\rho < 0$ and if $S(t)$ is still close to 1.

Remark 4. *Note that if $(\mathcal{E}, \mathcal{I})$ are two independent exponential random variables with parameters $\nu > 0$ and $\gamma > 0$, the equations of Definition 1 coincide with the law of large numbers limit of the Markovian SEIR epidemic model [4]. In this case, Proposition 2 agrees with what is already known about Markovian epidemic models. In particular, in the case $\rho > 0$,*

$$\lambda = (\rho + \gamma) \left(1 + \frac{\rho}{\nu}\right), \quad \mathbf{e} = \frac{\rho}{\nu + \rho}, \quad \mathbf{i} = \frac{\rho\nu}{(\rho + \gamma)(\nu + \rho)}, \quad \mathbf{r} = \frac{\nu\gamma}{(\rho + \nu)(\rho + \gamma)}.$$

The equivalent formulas for the SIR model (i.e., with $\mathcal{E} = 0$) can be obtained by letting $\nu \rightarrow \infty$, in which case we see that $\rho = \gamma(R_0 - 1)$, as in formula (1) in [2].

If, however, we assume that the variables \mathcal{E} and \mathcal{I} are constant and equal to (t_e, t_i) , the equations of Definition 1 can be seen as delay equations. In this case, Proposition 2 still applies, and the expectations in (4) and (5) can be omitted, leading to the relation

$$R_0 = \frac{\rho t_i}{e^{-\rho t_e}(1 - e^{-\rho t_i})}.$$

As a result, we see that, for the same growth rate ρ , the contact rate and the relative proportions of exposed, infectious and removed individuals vary depending on the distribution of the sojourn times $(\mathcal{E}, \mathcal{I})$, as illustrated on Figure 2.

Figure 2: Value of R_0 as a function of the growth rate ρ for different distributions of the exposed and infectious periods $(\mathcal{E}, \mathcal{I})$ with the same means. Four distributions are displayed: exponential (corresponding to the Markovian SEIR model), fixed, bimodal distribution mimicking Covid-19 (see Subsection 1.4) and Gamma distribution. All distributions have a mean exposed time of 4 days and a mean infectious time of 5.4 days, corresponding to a proportion of reported individuals of 0.7. The three dashed vertical lines show the growth rates of the Covid-19 epidemic in mainland France before and during the first lockdown, and during the second wave.

1.3 Changing the contact rate during the early phase of the epidemic

In France, as in many countries around the globe, strict lockdown measures were put into place at a fairly early stage of the COVID-19 epidemic. Assuming that, at this stage, the proportion of non-susceptible individuals remains negligible, the non-Markovian model reduces to

$$\begin{aligned}
 E(t) &= E(0)G_0^c(t) + \int_0^t \lambda(s)G^c(t-s)I(s)ds, \\
 I(t) &= I(0)F_1^c(t) + E(0)\Psi_0(t) + \int_0^t \lambda(s)\Psi(t-s)I(s)ds, \\
 R(t) &= R(0) + I(0)F_1(t) + E(0)\Phi_0(t) + \int_0^t \lambda(s)\Phi(t-s)I(s)ds.
 \end{aligned} \tag{8}$$

We shall assume that $E(0)$, $I(0)$ and $R(0)$ are as in (5) and that G_0^c , Ψ_0 , Φ_0 and F_1 are given by (6) with Θ an exponential random variable of parameter ρ_0 , and that the function $\lambda(\cdot)$

satisfies

$$\lambda(t) = \begin{cases} \lambda_0 & \text{if } t \leq t_L, \\ \lambda_1 & \text{if } t > t_L, \end{cases}$$

where λ_i satisfies (4) with $\rho = \rho_i$ for some $\rho_0 \neq \rho_1$ (note that ρ_1 may be negative). Then, by Proposition 2, for all $t \leq t_L$,

$$E(t) + I(t) + R(t) = e^{\rho_0 t}.$$

Moreover, we expect that, after a transitory period $\delta > 0$ following the change of contact rate, the linear system is well approximated by

$$E(t_L + \delta + t) \approx E(t_L + \delta)e^{\rho_1 t}, \quad I(t_L + \delta + t) \approx I(t_L + \delta)e^{\rho_1 t}, \quad (9)$$

for $t \geq 0$ (the interval δ corresponds to the time it takes for the system to “forget” the initial growth phase). Summing the three equations in (8), we see that,

$$E(t) + I(t) + R(t) = E(0) + I(0) + R(0) + \int_0^t \lambda(s)I(s)ds.$$

Let us write $EIR(t) = E(t) + I(t) + R(t)$. As a result, if (9) holds, then, for all $t \geq 0$,

$$\begin{aligned} EIR(t_L + \delta + t) &\approx EIR(t_L + \delta) + \lambda_1 \int_0^t I(t_L + \delta)e^{\rho_1 s} ds \\ &\approx EIR(t_L + \delta) + \lambda_1 I(t_L + \delta) \frac{e^{\rho_1 t} - 1}{\rho_1}. \end{aligned}$$

Moreover, by linearity, there exist positive constants $C_E(\delta)$, $C_I(\delta)$, $C_R(\delta)$, depending only on δ , ρ_0 , ρ_1 and the distribution of $(\mathcal{E}, \mathcal{I})$, such that

$$E(t_L + \delta) = C_E(\delta)e^{\rho_0 t_L}, \quad I(t_L + \delta) = C_I(\delta)e^{\rho_0 t_L}, \quad R(t_L + \delta) = C_R(\delta)e^{\rho_0 t_L}. \quad (10)$$

(We shall omit the dependence on δ in the following.) These constants are not known explicitly, but can be computed numerically by solving (8). Using this and Proposition 2, we obtain the following

$$EIR(t_L + \delta + t) \approx (C_E + C_I + C_R) e^{\rho_0 t_L} + \lambda_1 C_I e^{\rho_0 t_L} \frac{e^{\rho_1 t} - 1}{\rho_1}. \quad (11)$$

We shall use this later on to estimate the distributions of the delay between infection and hospital admission, hospital death and ICU admission.

1.4 Distribution of the sojourn times for Covid-19

The evolution of infectiousness for Covid-19 from the time of infection remains uncertain, but some early studies already provide constraints on the distribution of the sojourn times $(\mathcal{E}, \mathcal{I})$ for this disease. In particular, [17] ~~estimate~~ estimates that infectiousness starts as

early as 2.3 days before the onset of symptoms and declines within 7-8 days of symptom onset (see also [18]). Assuming that symptoms start on average 5.2 days after infection (as reported in [17]), we conclude that \mathcal{E} is between 2 and 4 days, and that \mathcal{I} is at least 3 days (if the infected individual is isolated shortly after the onset of symptoms) and no more than 10 to 12 days.

For the exposed time \mathcal{E} , we shall take a linear combination of the form $2 + 2X_1$, where X_1 is a symmetric Beta random variable with support in $[0, 1]$. We then assume that infected individuals are divided in two groups. Individuals in the first group, called the *reported* individuals, are isolated shortly after the onset of symptoms, either because they self-isolate at home or because they are admitted to a hospital, where we assume that they do not infect anyone else. For these individuals, we assume that $\mathcal{I} = 3 + X_2$, where X_2 is a symmetric Beta random variable with support in $[0, 1]$. By contrast, individuals in the second group remain infectious for much longer, either because they show very mild symptoms or no symptoms at all or because they fail to be isolated. For these *unreported* individuals, we assume that $\mathcal{I} = 8 + 4X_3$, where X_3 is a symmetric Beta random variable with support in $[0, 1]$.

Let p_R be the proportion of reported individuals and let Y be a Bernoulli random variable with parameter p_R , independent of X_2 and X_3 . Then we can write

$$\mathcal{I} = Y(3 + X_2) + (1 - Y)(8 + 4X_3).$$

We thus see that the distribution of \mathcal{I} is bimodal, with a first peak at around 3.5 days and a second one around 10 days.

It seems that asymptomatic individuals are less infectious than symptomatic ones (see [19]). We could have included in the model different values of infectivity depending upon the duration of the infectious period of each individual, using the theory developed in [20], but we lack quantitative information about the various levels of infectivity to make serious predictions, while the available data does not allow us to estimate many parameters. Note also that, as shown in [20], during the early phase of the epidemic, the product of the number of asymptomatic patients and their infectivity determines the evolution of the epidemic. Overestimating the infectivity leads to an underestimation of the number of cases, which has no significant impact on the ~~dynamic~~-dynamics of the epidemic during the early phase, but may lead to underestimation of the proportion of the population which has acquired immunity at the end of this early phase.

1.5 Estimating the contact rate prior to lockdown measures

In France, the Covid-19 epidemic started with a rapid exponential growth (of the number of cases, number of hospitalized patients, number of deaths), followed by a slowing down of the epidemic due to the lockdown measures put into place around March 16 [21], settling after a few weeks to an exponential decrease of the number of newly hospitalized patients and new deaths. This decrease continued after the partial lifting of travel restrictions on May 11, but came to a halt at the beginning of the summer, before hospital admissions, hospital deaths and ICU admissions started to increase again during the summer. This growth was stopped in mid-November, a few weeks after the second lockdown was put into place.

Since testing has been limited to a subset of symptomatic individuals and tests were performed at various intervals following symptom onset and with unequal intensity, the

number of reported positive cases might not be exactly proportional to the true number of infected individuals throughout the different stages of the epidemic (and more importantly the ratio of infected individuals to tested individuals may vary significantly between the different phases). Hospital deaths, however, have been reported daily from February 15. Moreover, from March 18 onwards, Santé Publique France has published daily reports of newly hospitalised patients, new admissions in intensive care units (ICU) and new deaths in each administrative *département*, [22].

Assuming that the distributions of the intervals between infection and hospital admission, admission in ICU and death do not vary over time, the growth rate of these quantities is necessarily identical to that of the cumulative number of infected individuals (see equation (13) below). As a result, if the distribution of the exposed and infectious periods (\mathcal{E}, \mathcal{I}) is known, we can infer the contact rate λ corresponding to a given growth rate ρ using (4).

Note that, since Santé Publique France only started publishing regional data on March 18, the growth rate for the initial phase in each region is inferred from the slope of the cumulative deaths during the first week of lockdown. Since the delay between infection and death in hospital is believed to be at least 10 days (*e.g.*, [23]), it is safe to assume that all the individuals who died during this period were infected before lockdown measures took effect (however, see Subsection 2.1 below for the particular case of the Grand Est and Hauts-de-France regions).

If moreover one knows the distribution of the interval \mathcal{D} between infection and death, for example, as well as the infection fatality ratio f (the probability that an individual eventually dies, given that he or she has been infected), it is possible to infer the state of the epidemic. Indeed, the cumulative number of deaths at time $t \geq 0$, denoted by $\Lambda_D(t)$, is then given by

$$\Lambda_D(t) = fN \int_0^t \left(-\frac{d\bar{S}(s)}{ds} \right) \mathbb{P}(\mathcal{D} < t - s) ds = fN \mathbb{E} [1 - \bar{S}(t - \mathcal{D})], \quad (12)$$

where N is the total population size (the expectation is taken with respect to the random variable \mathcal{D}). ~~During the~~

Note that since deceased individuals are no longer infectious, \mathcal{D} should be coupled with (\mathcal{E}, \mathcal{I}) so that, on the event that an individual dies, $\mathcal{E} + \mathcal{I} < \mathcal{D}$ (and similarly for individuals who are admitted to a hospital). As a result, individuals who die at time t (and who were infected at some time $t - \mathcal{D}$), should all be in the removed (R) state. The fact that $\bar{S}(t - \mathcal{D})$ appears in (12) instead of $\bar{R}(t - \mathcal{D})$ simply comes from the fact that \mathcal{D} denotes the delay between the time at which an individual *becomes infected*, and the time at which he/she dies.

Recall that, during the early phase of the epidemic,

$$N(1 - \bar{S}(t)) \approx E(t) + I(t) + R(t) = e^{\rho_0 t},$$

where t is the time elapsed since the (unknown) time of the start of the epidemic (*i.e.*, the theoretical time at which only one individual was infected in the population), and $E(t)$, $I(t)$ and $R(t)$ solve the linearized system (1). Thus, during this phase,

$$\Lambda_D(t) = f \mathbb{E} [\mathbf{1}_{\{\mathcal{D} \leq t\}} e^{-\rho_0 \mathcal{D}}] e^{\rho_0 t}. \quad (13)$$

It is thus possible to infer ρ_0 from the slope of $t \mapsto \log(\Lambda_D(t))$ during this initial phase and, using (4), one can then deduce the value of the contact rate before lockdown measures, λ_0 , (given that the distribution of $(\mathcal{E}, \mathcal{I})$ is known).

In order to estimate the start date of the epidemic and to be able to solve the system (1), we need information on f and \mathcal{D} . We shall see in Subsection 1.7 how to obtain information on the distribution of \mathcal{D} using what happens after a change in the contact rate.

1.6 Estimating the contact rate during lockdown measures

At the start of lockdown measures, the contact rate in the population dropped sharply over the course of a few days. As a result, the daily number of deaths in hospitals started to increase more slowly, before decreasing at a steady rate after approximately three weeks. In addition, the number of new hospital admissions and ICU admissions also decreased at a steady rate after a (shorter) transition period.

Assuming that, at this point, the proportion of susceptible individuals in the overall population remains close to 1, this corresponds to the situation described by (8) with $\rho_1 < 0$. Using (11) and (12) and choosing t large enough (such that $\mathbb{P}(\mathcal{D} > t)$ is small), we have

$$\Lambda_D(t_L + \delta + t) \approx f(C_E + C_I + C_R) e^{\rho_0 t_L} + \frac{f \lambda_1 C_I e^{\rho_0 t_L}}{\rho_1} (\mathbb{E}[e^{-\rho_1 \mathcal{D}}] e^{\rho_1 t} - 1). \quad (14)$$

In particular, there exists a constant $C > 0$ such that

$$\Lambda'_D(t_L + \delta + t) = C e^{\rho_1 t},$$

and the same holds for hospital admissions (with a different constant C). As a result, we can estimate ρ_1 from the cumulative number of hospital admissions and hospital deaths, and deduce the value of the contact rate during lockdown using (4).

Here and in what follows, we assume that the distribution of the sojourn times $(\mathcal{E}, \mathcal{I})$ is unaffected by the various containment measures put into place during the course of the epidemic. This may not be true, especially if contact tracing and massive testing are implemented at a sufficient scale in the population, but the extent of such measures in France, at least during the early stages of the Covid-19 epidemic, remained very limited.

1.7 Estimating the distribution of the delay between infection and hospital death or hospital admission

We have seen above that (14) can be used to determine the contact rate during lockdown measure, but in fact we are able to estimate all the constants appearing in (14) by fitting an exponential curve to the cumulative number of hospital deaths. More precisely, we can estimate the values of both

$$A = \frac{f \lambda_1 C_I e^{\rho_0 t_L}}{\rho_1} \mathbb{E}[e^{-\rho_1 \mathcal{D}}], \quad \text{and} \quad B = f(C_E + C_I + C_R) e^{\rho_0 t_L} - \frac{f \lambda_1 C_I e^{\rho_0 t_L}}{\rho_1}.$$

Adding relation (13) with $t = t_L$ (and assuming that $\mathcal{D} < t_L$ a.s.), we also have

$$\Lambda_D(t_L) = f \mathbb{E}[e^{-\rho_0 \mathcal{D}}] e^{\rho_0 t_L}. \quad (15)$$

In addition to f , the only unknowns here are $\mathbb{E}[e^{-\rho_0 \mathcal{D}}]$, $\mathbb{E}[e^{-\rho_1 \mathcal{D}}]$ and t_L . Substituting $f e^{\rho_0 t_L} = \Lambda_D(t_L) \mathbb{E}[e^{-\rho_0 \mathcal{D}}]^{-1}$ in the first two equations, we obtain

$$\begin{aligned}\mathbb{E}[e^{-\rho_0 \mathcal{D}}] &= \frac{\Lambda_D(t_L)}{B} \left[(C_E + C_I + C_R) - \frac{\lambda_1 C_I}{\rho_L} \right], \\ \mathbb{E}[e^{-\rho_1 \mathcal{D}}] &= \frac{A}{B} \left[\frac{\rho_1}{\lambda_1 C_I} (C_E + C_I + C_R) - 1 \right].\end{aligned}\tag{16}$$

These two identities do not fully characterize the distribution of \mathcal{D} , but if we assume that this distribution belongs to some two-parameter family, it is possible to estimate the corresponding parameters. In the following, we shall assume that \mathcal{D} follows a Gamma distribution with parameters (θ, k) . Then, for all $\rho > -\theta^{-1}$,

$$\mathbb{E}[e^{-\rho \mathcal{D}}] = (1 + \rho \theta)^{-k}.$$

The parameters θ and k can then be computed numerically by inverting (16). Then, using (15), we can deduce the value of t_L , the time elapsed since the beginning of the epidemic at the start of lockdown. We can further note that one does not need to know the value of f in order to compute (16), and to estimate the distribution of \mathcal{D} . This is fortunate, as the parameter f is difficult to estimate precisely.

Note that the Gamma distribution has support in $[0, \infty)$ and so does not satisfy the requirement that $(\mathcal{E}, \mathcal{I})$ and \mathcal{D} be coupled so that, on the event that an individual dies, $\mathcal{E} + \mathcal{I} < \mathcal{D}$. However, we shall see below that the average delay between infection and hospital admission (hence hospital deaths) is large enough that this inequality can be achieved with high probability. Given the low percentage of individuals who are admitted to hospitals (of the order of 2-3%), this does not significantly affect our results.

We proceed in the same way for the delay between infection and hospital admission and ICU admission. Once we have estimated the parameters of the delay distribution for each of these events, we can compute the probability of hospital admission and ICU admission as follows.

1.8 Estimating the probability of hospital and ICU admission

Note that if p_H is the probability that a given infected individual is admitted to a hospital at some point, and if \mathcal{D}_H is the interval between the time of infection and hospital admission, then, the cumulative number of hospitalized patients at time $t \geq 0$, denoted by $\Lambda_H(t)$, is given by

$$\Lambda_H(t) = p_H N \mathbb{E} [1 - \bar{S}(t - \mathcal{D}_H)].$$

Hence, during the early phase of the epidemic,

$$\Lambda_H(t) = p_H \mathbb{E} [\mathbf{1}_{\{\mathcal{D}_H \leq t\}} e^{-\rho_0 \mathcal{D}_H}] e^{\rho_0 t}.$$

As a result, one can obtain the probability of being admitted to hospital as a function of f through the relation,

$$p_H = f \frac{\Lambda_H(t) \mathbb{E} [e^{-\rho_0 \mathcal{D}}]}{\Lambda_D(t) \mathbb{E} [e^{-\rho_0 \mathcal{D}_H}]},\tag{17}$$

for any time $t \in [0, t_L]$ for which $\mathcal{D} < t$ and $\mathcal{D}_H < t$ almost surely, where t_L is the time at which lockdown measures are put into place.

1.9 Estimating the contact rate after the easing of the first lockdown measures

The easing of lockdown restrictions was organized in different phases. On May 11, people were allowed to leave their homes, shops started to reopen and schools progressively welcomed pupils again. On June 2, bars and restaurants reopened in most of the country, and on June 22, all schools reopened, along with cinemas, and most activities resumed, although sanitary measures continued to be enforced (*e.g.*, wearing a mask remained mandatory in many public places including trains and public transports).

As we shall see below, at the end of the first lockdown period, the proportion of susceptible individuals in the population had already dropped by 5 to 10%, at least in Île de France and the Grand Est and Hauts-de-France regions. As a result, the approximation $\bar{S}(t) \approx 1$ may not be valid at this point, preventing us from directly applying (4). However, since the daily new infections remain at low levels over short time periods, the value of $\bar{S}(t)$ does not vary much during the period over which we estimate the new growth rates ρ_2, ρ_3, ρ_4 . It follows that we can replace λ by ~~$\bar{S}(t_E)\lambda$~~ the effective rate $\lambda_e = \bar{S}(t_E)\lambda$ in (4), where $\bar{S}(t_E)$ is the proportion of susceptible individuals at the end of the lockdown period, ~~and~~. We hence deduce the corrected value of the effective contact rate during each period following the easing of lockdown restrictions. These effective rates correspond to effective reproduction numbers $R_e = \bar{S}(t_E)R_0$, which reduce the original “basic reproduction number” R_0 .

1.10 Estimating the state of the epidemic

Once we have estimated the contact rates prior to lockdown (λ_0), during lockdown (λ_1) and after lockdown ($\lambda_i, i \geq 2$), as well as the interval between the (theoretical) index case and the start of lockdown t_L , we are in a position to compute the state of the epidemic. To do this we numerically solve the equations of Definition 1 with

$$\lambda(s) = \begin{cases} \lambda_0 & \text{if } s < t_L, \\ \lambda_1 & \text{if } t_L \leq s < t_E, \\ \lambda_i & \text{if } t_i \leq s \leq t_{i+1}, \text{ for } i \geq 2, \end{cases}$$

where t_L is the time at which lockdown measures are implemented and t_E is the time at which these measures are eased. We also take $t_2 = t_E$ and t_i for $i \geq 3$ denotes the subsequent times of changes of the contact rate in the population. For the initial condition, using Proposition 2, we choose

$$\bar{S}(0) = 1 - \frac{1}{N}, \quad \bar{E}(0) = \frac{\mathbf{e}}{N}, \quad \bar{I}(0) = \frac{\mathbf{i}}{N}, \quad \bar{R}(0) = \frac{\mathbf{r}}{N},$$

with $\mathbf{e}, \mathbf{i}, \mathbf{r}$ as in (1) and $G_0^c, \Psi_0, \Phi_0, F_1$ as in (6), choosing ρ equal to ρ_0 , the growth rate prior to lockdown in (1) and as the parameter of the exponential random variable Θ in (6). In this way, by Proposition 2, $1 - \bar{S}(t) = \frac{1}{N}e^{\rho_0 t}$ for all $t \leq t_L$, as expected.

It might seem counter-intuitive to start the epidemic with some fraction of removed individuals ($\bar{R}(0) > 0$), even more so as we assume that only one individual is not susceptible at this time. One should keep in mind that we are not trying to estimate the *true* initial state of the epidemic; we merely find a suitable initial condition so that the observed exponential growth prior to lockdown measures fits the observed data. Starting our model with exactly one infectious individual at time 0 would lead to the same exponential growth behaviour, but after a short transitory period which would shift our predictions.

2 Results

We present the results of our estimations in four different French regions: Île de France, Grand Est, Provence-Alpes-Côte d’Azur (PACA) and Auvergne-Rhône-Alpes. The epidemic was first detected in the Grand Est region, after which it quickly spread to the Paris region, and then the whole country. As a result, some regions were much more affected than others during the first wave in March and April, and this can be seen directly in the hospital data from Santé Publique France [22].

2.1 Growth rates of the epidemic

We measured the growth rate of the cumulative number of infected individuals during the early phase of the epidemic by fitting an exponential curve to the cumulative number of deaths in each patch, between March 19 and March 26 (earlier data was only available at the national level). Despite the fact that very strict lockdown measures were already effective at the time, hospital deaths continued to increase exponentially until at least March 26, mainly due to the fact that infected individuals who died of Covid-19 during this period had been infected before national lockdown, hence during the exponential growth phase.

We can see that the initial growth rate in the Grand Est region is the slowest, despite the fact that this is where the epidemic first started. This can be attributed to the fact that some public gatherings were banned in this region about 10 days before the national lockdown. Hence this might not reflect the growth rate of the epidemic from its very start, which might be closer to the one measured at the national level (doubling time of 2.5 days, using hospital deaths data from March 1).

After a few weeks of lockdown, daily hospital admissions, hospital deaths and ICU admissions started to decline, again exponentially, at a steady rate in each region. We observe a drop in the number of daily admissions and deaths every ~~week-end~~~~weekend~~, which was systematically compensated at the start of the following week, indicating reporting delays. We thus fitted *simultaneously* three exponential curves to the *cumulative* number of hospital admissions, hospital deaths and ICU admissions between April 3 and May 11 (starting only on April 13 for ICU admissions and deaths). The obtained values are given in the second line of Table 1. Note that the halving time during lockdown was much longer than the doubling time prior to lockdown, explaining why it had to remain in place for several months in order to significantly reduce the number of infectious individuals.

Using the same method, we estimated the growth rate of the cumulative number of infected individuals in subsequent stages of the epidemic (Figure 3). The growth rate of the

Region	Île de France	Grand Est	PACA	Auvergne-Rhône-Alpes
Doubling time before lockdown (from hospital deaths)	2.1 days	3 days	2.8 days	2 days
Halving time during first lockdown	11.6 days	12.4 days	15.9 days	13.9 days
Doubling time during the second wave	21.4 days	36.9, then 7 days	11.5, then 9.2 days	14 days
Halving time during second lockdown	14.9 days	17.7 days	16.2 days	14.1 days

Table 1: Doubling and halving times measured during the main phases of the epidemic in each of the regions presented here. Doubling times prior to lockdown are measured using hospital deaths between March 19 and March 26, but are consistent with what is obtained for the whole country (2.5 days) using data from March 1st.

epidemic remained negative until June, and hospital admissions, followed by ICU admissions and deaths, started to increase again during the summer. This increase continued during the fall in every region, but much more slowly than before the first lockdown, probably due to all the measures put into place since then (mandatory masks in many public places, more people working from home, etc.). We can also note that the growth rate varies between regions, and that it is higher in regions that were less hit by the first wave. This can be explained by the fact that, since fewer people have been infected during the first wave in such places, the overall immunity level is also lower, and hence the epidemic spreads faster. The second wave was then stopped after the start of the second national lockdown on October 30. The latest data also shows a plateau, and even a slight increase in some regions of the hospital admissions over the last weeks of December. If confirmed, this trend would mean that a third wave might hit again very early in 2021.

2.2 Reproduction numbers and the effect of lockdown measures

Using the relation (4) and our assumptions on the distribution of $(\mathcal{E}, \mathcal{I})$ from Subsection 1.4, we can estimate the values of the reproduction number R_0 in each region during each phase of the epidemic. Since the growth rates prior to lockdown and during lockdown are relatively uniform across the different regions, we use the same value for all patches. For the second wave, we compute R_0 with the two extreme values (doubling every 11.5 days or every 21.4 days).

The proportion of reported individuals affects both the expectation of the infectious period $\mathbb{E}[\mathcal{I}]$ and the whole distribution of $(\mathcal{E}, \mathcal{I})$. Considering Corollary 3, we thus observe that the estimated value of the reproduction number depends on the proportion of reported individuals, and more generally on the whole distribution of $(\mathcal{E}, \mathcal{I})$, as shown on Table 2.

The relatively large values (around 6) obtained for small proportions of reported individuals suggest that this proportion is closer to 0.8 than to 0.2. Moreover, the values we obtain,

Figure 3: Growth rates of the cumulative number of infected individuals before, during and after lockdown in France, deduced from the number of hospital admissions, ICU admissions and hospital deaths in each patch. In each instance, an exponential curve was fitted simultaneously to each cumulative number of hospital admissions, ICU admissions and deaths. The last value in each patch is deduced from the cumulative number of hospital admissions, ICU admissions and hospital deaths during the last two weeks of June.

even for large proportions of reported individuals, are sometimes larger than other estimates in the literature (Table 3), although we note that our estimate of R_0 fits well the results in [24]. We can attribute this to two things. First, as shown in Figure 2, the Markovian SEIR model tends to slightly underestimate R_0 in the two regimes we are interested in (growth in the early phase and decrease under lockdown). Second, some studies do not take into account the exposed period \mathcal{E} in their models. However, Corollary 3 shows that neglecting the exposed period leads to underestimating of the reproduction number R_0 , here by a factor of 0.76 for each day of exposed period ($e^{-\rho} = 0.76$ with $\rho = 0.27$). Finally, uncertainty about the infectious period \mathcal{I} also affects the estimates of the reproduction number. More details are given on this in Appendix D, using a sensitivity analysis.

Proportion of reported individuals	R_0 prior to lockdown	R_0 during the first lockdown	R_0 during the second wave
0	6.3	0.66	1.28 - 1.56
0.2	6	0.67	1.27 - 1.55
0.8	4.2	0.73	1.21 - 1.42
1	3.4	0.79	1.16 - 1.32

Table 2: Values of R_0 during the three main phases of the epidemic for different proportions of reported individuals, using the sojourn time distribution from Subsection 1.4.

Country	R_0 prior to lockdown	R_0 during lockdown	Accounting for exposed period	Reference
France	2.9 - 3.35	0.55 - 0.72	implicit	[5]
France	3.1 - 3.3	0.47	no	[25]
France	3.3	0.5	yes	[23]
France	3.4 ± 0.1	0.65 ± 0.04	yes	[26]
China	4.7 - 6.6	/	yes	[24]

Table 3: Estimates of the reproduction number in the literature

2.3 State of the epidemic and acquired immunity at the population level

Using (13) and Subsection 1.10, we can estimate the state of the epidemic in each region, given the infection fatality ratio f and the distribution of the delay \mathcal{D} . Unfortunately, these two quantities are notoriously hard to measure during the early stage of the epidemic.

The infection fatality ratio for the Covid-19 epidemic in France has been estimated in at least two studies [25, 23]. Both studies found a fatality ratio of 0.5%, with significant variation across age classes in [23]. These studies only account for hospital deaths, as we do, even though a significant number of deaths take place outside hospitals, mainly in nursing homes. Hence this ratio has to be corrected (roughly by a factor 1.6 [25]) to obtain the true infection fatality ratio of Covid-19 (and to obtain correct predictions for the expected number of deaths). Nevertheless, since we use hospital deaths to calibrate our SEIR model, we shall use the infection fatality ratio estimated by [23, 25] when using (13).

We can at least bound this ratio from below using the observed excess mortality in some regions. For example, in Lombardy (Italy), 16,973 people died of Covid-19, for a total population of 10 million, showing that this ratio is at least 0.17% (or 0.11% for hospital deaths). Another estimate of this ratio was obtained in a small town in Germany, where around 1,000 individuals were tested, out of which 15.5% tested positive and 7 people died, yielding an infection fatality ratio of 0.37% [27] (corresponding to 0.23% for hospital deaths).

On the other hand, the infection fatality ratio can be bounded from above by the apparent death rate, that is to say, the ratio between Covid-19 related deaths and declared positive cases, at least while the epidemic seems to be receding, as is the case in France in June 2020, with less than 50 deaths per day. Taking only hospital deaths into account, this suggests that f is no more than 12%. Other countries have much lower apparent death rates, *e.g.*, South

Korea (2.3%), Germany (4.6%). These discrepancies can mostly be attributed to differences in testing capacities and to a lesser extent to differences in hospital capacities.

Using the method described in Subsection 1.7, we estimated the parameters (k, θ) for the delay between infection and each of the three possible events (hospital admission, ICU admission and hospital death). The means and standard deviations of the distributions we obtained are given in Table 4 and the distributions obtained in Île-de-France are shown on Figure 4.

Region	Hospital admission	ICU admission	Hospital death
Île-de-France	13.6 ± 6.3 days	11.1 ± 5.2 days	20.1 ± 7.6 days
Grand Est	11.7 ± 5.5 days	9.3 ± 2.6 days	16.2 ± 7.5 days
PACA	12.7 ± 6.7 days	13.9 ± 5.3	23.7 ± 9 days
Auvergne-Rhône-Alpes	11.4 ± 4 days	8.8 ± 1.7 days	20 ± 8.4 days

Table 4: Mean and standard deviations of the delays between infection and hospital admission, ICU admission and hospital deaths obtained from (16).

The delays appear to be slightly shorter in the Grand Est region than everywhere else. This can be a consequence of the fact that the growth rate measured during the first weeks of lockdown does not reflect the true initial growth rate of the epidemic in this region, as noted in Subsection 2.1.

Figure 4: Probability density functions of the delays between infection and hospital admission (blue), ICU admission (green) and Hospital death (orange) estimated in Île-de-France. Means are indicated in dashed vertical lines.

We estimated the probability of being admitted to hospital given infection. This obviously depends on the infection fatality ratio, but, on the other hand, the death to admission ratio (*i.e.*, the probability of dying given that one is admitted to a hospital) is constant.

We estimated that the death to admission ratio was 0.19 in Île de France, 0.20 in the Grand Est region, 0.14 in PACA and 0.17 in Auvergne-Rhône-Alpes. Note that we consider that these are constant throughout the epidemic, which may not be the case, especially if hospitals become overwhelmed by the influx of patients. This could for example explain the discrepancy between the death to admission ratios in the different regions, which are computed using the data from the first weeks of lockdown.

Using the resulting estimates on the delay distributions and the probabilities of hospital admission and ICU admission, we can simulate the deterministic SEIR model from Definition 1 with the contact rates estimated in Figure 3, and compute the number of hospital admissions, ICU admissions and hospital deaths in the model. The result is represented in Figure 5. The fact that our delay distributions satisfy (16) ensures that our model reproduces the initial growth before March 16 and the exponential decrease during the first lockdown.

Interestingly, in two regions (PACA and Grand Est), we are not able to fit the decrease during the second lockdown without an intermediary step with a slower increase around the time at which the lockdown started (see Figure S2). This can be explained by the fact that, two weeks before the second lockdown started, many large cities declared a nightly curfew from 9pm to 6am.

Figure 6 shows the predicted levels of immunity (*i.e.*, the proportion of infected individuals) up to December 31 in Île-de-France for three values of the infection fatality ratio f , using the delay distributions estimated above. As expected, higher values of this ratio lead to lower predicted levels of immunity. Note that underestimating \mathcal{D} has the same effect as overestimating the infection fatality ratio, as can be noted from (13). As a result it is equally important to estimate f and the distribution of \mathcal{D} , a fact which is too easily overlooked.

For the fatality ratio estimated by [25, 23] ($f = 0.5\%$), we estimate that around 20% of the population has been infected by the end of December in Île de France. In the Grand Est region, this proportion is 25%, while in PACA, only 15% seem to have been infected, and 17% in Auvergne-Rhône-Alpes. As expected, regions that were infected later during the first wave now have lower levels of immunity than the two regions where the epidemic started early. As a consequence, during a possible third epidemic wave, regions with relatively lower cumulative number of death are more at risk of seeing a sharper increase in the number of cases.

3 Discussion

One of our goals in this article was to study, in the context of an ongoing epidemic, which knowledge about the parameters of a given model can be extracted from a certain type of data, and what needs to be estimated separately. In addition, our aim is to stress how the choice of the model (for instance choosing a Markovian or a non-Markovian model) affects the predictions one can make based on the data.

We have exposed a method to fit a very general epidemic model to hospital data, using as many analytical tools as we could. As a result, the procedure outlined in this paper **allows**, summed up in Appendix B, **allows one** to estimate and monitor the state of an ongoing epidemic all along its course. Indeed, the computing power and time required to fit the model, even with many changes in the contact rate, remains very limited. The main reason

Figure 5: Model predictions and hospital data throughout the epidemic in four regions. In each region, we confront the daily hospital admissions, ICU admissions and hospital deaths due to Covid-19 to the predictions of our model. The model starts with a single infected individual at a different date in each region, and ends on December 31. The value used for the infection fatality ratio f was 0.5%, following [25], and the proportion of reported individuals was 0.8.

for this is that we do not need to explore the space of parameters and simulate the SEIR model for many sets of parameters. Instead we compute each parameter step by step, using only the information in the exponential fits displayed in Figure 3.

As exposed in Remark 4, knowing the distribution of the sojourn times in each compartment $(\mathcal{E}, \mathcal{I})$ is crucial to deduce the value of R_0 from the initial growth rate of the epidemic. Without knowledge on these sojourn times, the reproduction number can only be estimated using contact tracing, which is both difficult and costly. As shown in Figure 2, Markovian epidemic models (but not only) can in some regime underestimate the value of R_0 for a given initial growth rate of the epidemic. If this is the case, then they will underestimate the final proportion of infected individuals in the population in the absence of containment measures, and thus the predicted death toll of the epidemic.

~~Comparison of the daily numbers of newly infected individuals in a Markovian SEIR~~

Figure 6: Predicted level of immunity in Île-de-France as a function of the infection fatality ratio f . All the simulations use $p_R = 0.8$. As expected, larger infection fatality ratios lead to lower predicted immunity, since fewer people need to have been infected to ~~reproduce~~ reproduce the observed number of deaths.

~~epidemic model and in a non-Markovian SEIR epidemic model. The two models have the same initial growth rate and the same mean exposed and infectious periods. The population size was taken to be 10,000. In addition, the Markovian model also seems to underestimate the height of the peak of the epidemic wave, *i.e.* the maximum daily new infections during the course of the epidemic, as shown on Figure ??, where the non-Markovian model predicts 100 more new patients in a single day than the Markovian model for a total population of only 10,000. We can attempt a heuristic explanation of this. In the Markovian model, the daily number of infections decreases just after the proportion of susceptible individuals reaches $1/R_0$, while in the non-Markovian model, one has to wait a little more in order to see this decrease, due to memory effects. Since the height of this peak determines whether or not hospitals are overwhelmed, the consequences of underestimating it can be severe.~~

On the other hand, one needs additional information on the infection fatality ratio (*i.e.*, the proportion of infected individuals who eventually die) and on the distribution of the delay between infection and death in order to estimate the true state of the epidemic at any given time and thus the level of acquired immunity in the population. We have shown that some information on this delay distribution can be retrieved from the time series of hospital deaths caused by the disease, but this information is partly contingent on the model used to estimate the state of the epidemic (that is, on the distribution of $(\mathcal{E}, \mathcal{I})$).

More crucially, one has to wait for some sort of containment measures to take effect in order to estimate these delays, hence this cannot be done during the initial growth phase of the epidemic. Since Markovian models tend to underestimate the inertia of the epidemic after containment measures are put into place (*i.e.*, they tend to decrease exponentially too soon after a change in the contact rate), using these will lead to an underestimation

of the length and height of the epidemic *plateau* reached before the decreasing phase. This can be seen in the slow decrease in hospital admissions and deaths even after 3 weeks of strict lockdown, something which the Markovian ODE model would not have predicted, but that the non-Markovian SEIR model reproduces naturally. ~~One way to force the Markovian model to display this kind of behavior~~

This lack of inertia in ODE models results from the lack of memory property of the exponential distribution: the amount of time an individual already in the E or I compartment stays in this compartment does not depend on the time they have already spent in this compartment, contrary to other more realistic models such as the one considered in the present work. Of course it remains possible to choose the joint distribution of $(\mathcal{E}, \mathcal{I})$ such that the model with memory decreases more rapidly than the ODE model after the start of lockdown measures (for example by choosing Gamma distributions with $k < 1$ for \mathcal{E} and \mathcal{I} , see Appendix C), but such distributions seem even less realistic than the exponential distribution for infectious disease modelling.

As a result, we expect that using ODE models to fit Covid-19 hospital data forces one to compensate for this in some way. One way to do this would be to make the change in the contact rate more gradual, but we do not believe that this would reflect what took place in France. Alternatively, one could artificially increase the mean exposed time in the model, which results in a longer delay between the change in the contact rate and the change of slope in the daily new infections. However this would significantly affect the estimates of R_0 , as can be seen from (4). Another way to force the Markovian-ODE model to fit this behavior would be to ~~overestimate-increase~~ the delay between infection and hospital death, ~~which would artificially increase the Markovian model's inertia,~~ but this would also lead to an overestimation of t_L , and ~~hence~~ of the acquired population immunity.

One thing that we can note is that the kind of data studied in this article carries very little information on the actual distribution of $(\mathcal{E}, \mathcal{I})$, at least during the early stages of an epidemic. External information on the true delay distribution between infection and death, or infection and hospital admission, *etc.*, can help to put constraints on the model, yielding more accurate predictions of the state of the epidemic.

The most obvious simplification in our analysis is the fact that we neglect the age structure in our population. This structure particularly affects the dynamics of Covid-19 since the infection fatality ratio varies significantly with age [23]. Here, we have only used the global infection fatality ratio computed in [23], but a more complex model could be fitted to the age-stratified data provided by Santé Publique France. Moreover the distribution of the exposed and infectious periods is likely to depend on the age of the infected individual, something which our model fails to consider. Our motivation in neglecting age structure was to keep the model mathematically tractable and to avoid ending up with too many parameters. In order to fully exploit the age-stratified data, one would further need external information on the matrix of contact rates between age-classes, something which is likely to have changed in complex ways as a result of all the containment policies aimed at limiting the spread of the virus.

Note that an analysis similar to the one conducted in this paper can be carried out for a model where the infectivity of each individual is a random function of the time since infection, those random functions for the various individuals being i.i.d., see [20]. This would yield a model which could account for the variability, both in time and between individuals, of the

infectivity of each infected individual, something which is apparent for Covid-19 for example in [17]. Applying this type of models to the ongoing Covid-19 epidemic, and including age-structure, will be the subject of future work.

Conflicts of interest The authors declare no conflict of interest.

Code availability The code used to simulate the SEIR model and to analyse the data is made available at https://github.com/rforien/Fit_Covid19_nonMarkovian.git

References

- [1] Guodong Pang and Etienne Pardoux, 2020. Functional Limit Theorems for Non-Markovian Epidemic Models. *arXiv:2003.03249 [math.PR]*.
- [2] Tom Britton, 2020. Basic estimation-prediction techniques for Covid-19, and a prediction for Stockholm. *medRxiv*.
- [3] Tom Britton, 2020. Basic prediction methodology for covid-19: Estimation and sensitivity considerations. *medRxiv*.
- [4] Tom Britton and Etienne Pardoux, 2019. Stochastic epidemics in a homogeneous community. *Stochastic Epidemic Models with Inference (T. Britton and E. Pardoux eds). Part I. Lecture Notes in Math. 2255*, pages 1–120.
- [5] Mircea T. Sofonea, Bastien Reyné, Baptiste Elie, Ramsès Djidjou-Demasse, Christian Selinger, Yannis Michalakis, and Samuel Alizon, 2020. Epidemiological monitoring and control perspectives: Application of a parsimonious modelling framework to the COVID-19 dynamics in France. *preprint hal-02619546*. doi: 10.1101/2020.05.22.20110593.
- [6] Zoltan Fodor, Sandor D. Katz, and Tamas G. Kovacs, 2020. Why integral equations should be used instead of differential equations to describe the dynamics of epidemics. *arXiv:2004.07208*.
- [7] William Ogilvy Kermack and Anderson G. McKendrick, 1927. A contribution to the mathematical theory of epidemics. *Proceedings of the Royal Society of London. Series A.*, 115(772):700–721. doi: 10.1098/rspa.1927.0118.
- [8] Stéphane Gaubert, Marianne Akian, Xavier Allamigeon, Marin Boyet, Baptiste Colin, Théotime Grohens, Laurent Massoulié, David Parsons, Frederic Adnet, and Érick Chanzy, 2020. Understanding and monitoring the evolution of the Covid-19 epidemic from medical emergency calls: The example of the Paris area. *Comptes Rendus. Mathématique*, 358(7):843–875. doi: 10.5802/crmath.99.
- [9] Félix Foutel-Rodier, François Blanquart, Philibert Courau, Peter Czuppon, Jean-Jil Duchamps, Jasmine Gamblin, Élise Kerdoncuff, Rob Kulathinal, Léo Régnier, and

- Laura Vuduc, 2020. From individual-based epidemic models to McKendrick-von Foerster PDEs: A guide to modeling and inferring COVID-19 dynamics. *arXiv:2007.09622 [q-bio.PE]*.
- [10] Dorothy Anderson and Ray Watson, 1980. On the spread of a disease with gamma distributed latent and infectious periods. *Biometrika*, 67(1):191–198. doi: 10.1093/biomet/67.1.191.
- [11] Ping Yan and Zhilan Feng, 2010. Variability order of the latent and the infectious periods in a deterministic SEIR epidemic model and evaluation of control effectiveness. *Mathematical biosciences*, 224(1):43–52. doi: 10.1016/j.mbs.2009.12.007.
- [12] Fred Brauer, Carlos Castillo-Chavez, and Zhilan Feng. *Mathematical Models in Epidemiology*, volume 32. Springer, 2019. ISBN 978-1-4939-9826-5.
- [13] David Alexander Madore, 2020. Exact solutions and analysis of an SIR variant with constant-time recovery. *preprint hal-02537265*.
- [14] Jacco Wallinga and Marc Lipsitch, 2007. How generation intervals shape the relationship between growth rates and reproductive numbers. *Proceedings of the Royal Society B: Biological Sciences*, 274(1609):599–604. doi: 10.1098/rspb.2006.3754.
- [15] Fred Brauer, 2005. The Kermack–McKendrick epidemic model revisited. *Mathematical biosciences*, 198(2):119–131. doi: 10.1016/j.mbs.2005.07.006.
- [16] Zhihua Liu, Pierre Magal, Ousmane Seydi, and Glenn Webb, 2020. Predicting the cumulative number of cases for the COVID-19 epidemic in China from early data. *Mathematical biosciences and engineering : MBE*, 17(4):3040–3051. doi: 10.3934/mbe.2020172.
- [17] Xi He, Eric HY Lau, Peng Wu, Xilong Deng, Jian Wang, Xinxin Hao, Yiu Chung Lau, Jessica Y. Wong, Yujuan Guan, and Xinghua Tan, 2020. Temporal dynamics in viral shedding and transmissibility of COVID-19. *Nature medicine*, 26(5):672–675. doi: 10.1038/s41591-020-0869-5.
- [18] Peter Ashcroft, Jana S. Huisman, Sonja Lehtinen, Judith A. Bouman, Christian L. Althaus, Roland R. Regoes, and Sebastian Bonhoeffer, July 2020. COVID-19 infectivity profile correction. *arXiv:2007.06602 [q-bio, stat]*.
- [19] Pascal Cossart, Olivier Schwartz, Patrick Couvreur, Frédéric Tangy, Dominique Costagliola, Olivier Faugeras, and Arnaud Fontanet. COVID-19 - SÉANCE EXCEPTIONNELLE DE L'ACADÉMIE DES SCIENCES.
- [20] Raphaël Forien, Guodong Pang, and Étienne Pardoux, 2020. Epidemic models with varying infectivity. *arXiv:2006.15377 [math.PR]*.
- [21] Simon Cauchemez, Cécile Tran Kiem, Juliette Paireau, Patrick Rolland, and Arnaud Fontanet, 2020. Lockdown impact on COVID-19 epidemics in regions across metropolitan France. *The Lancet*, 396(10257):1068–1069. ISSN 0140-6736. doi: 10.1016/S0140-6736(20)32034-1.

- [22] Santé Publique France. Données hospitalières relatives à l'épidémie de COVID-19, 2020. URL <https://www.data.gouv.fr/fr/datasets/donnees-hospitalieres-relatives-a-lepidemie-de-covid-19/>.
- [23] Henrik Salje, Cécile Tran Kiem, Noémie Lefrancq, Noémie Courtejoie, Paolo Bosetti, Juliette Paireau, Alessio Andronico, Nathanaël Hozé, Jehanne Richet, and Claire-Lise Dubost, 2020. Estimating the burden of SARS-CoV-2 in France. *Science*, 396(6500). doi: 10.1126/science.abc3517.
- [24] Steven Sanche, Yen Ting Lin, Chonggang Xu, Ethan Romero-Severson, Nicolas W. Hengartner, and Ruian Ke, 2020. The novel coronavirus, 2019-nCoV, is highly contagious and more infectious than initially estimated. *arXiv:2002.03268 [q-bio.PE]*.
- [25] Lionel Roques, Etienne K. Klein, Julien Papaix, Antoine Sar, and Samuel Soubeyrand, 2020. Using early data to estimate the actual infection fatality ratio from COVID-19 in France. *Biology*, 9(5):97. doi: 10.3390/biology9050097.
- [26] Gary A. Mamon, 2020. Fit of French COVID-19 hospital data with different evolutionary models: Regional measures of R_0 before and during lockdown. *arXiv:2005.06552*.
- [27] Hendrik Streeck, Bianca Schulte, Beate Kuemmerer, Enrico Richter, Tobias Höller, Christine Fuhrmann, Eva Bartok, Ramona Dolscheid, Moritz Berger, and Lukas Wessendorf, 2020. Infection fatality rate of SARS-CoV2 in a super-spreading event in Germany. *Nature communications*, 11(1):1–12. doi: 10.1038/s41467-020-19509-y.

A Proof of Proposition 2

We start with the case $\rho > 0$. Replacing $(E(t), I(t), R(t))$ in (2) by their expressions in (3) yields the following equations

$$\begin{aligned} \mathbf{e}e^{\rho t} &= \lambda \mathbf{i} \int_{-\infty}^t G^c(t-s)e^{\rho s} ds \\ \mathbf{i}e^{\rho t} &= \lambda \mathbf{i} \int_{-\infty}^t \Psi(t-s)e^{\rho s} ds \\ \mathbf{r}e^{\rho t} &= \lambda \mathbf{i} \int_{-\infty}^t \Phi(t-s)e^{\rho s} ds. \end{aligned}$$

The second equation translates into

$$\begin{aligned} \lambda^{-1} &= \int_0^{\infty} \mathbb{P}(\mathcal{E} \leq s < \mathcal{E} + \mathcal{I}) e^{-\rho s} ds \\ &= \mathbb{E} \left[\int_{\mathcal{E}}^{\mathcal{E} + \mathcal{I}} e^{-\rho s} ds \right] \\ &= \frac{\mathbb{E} [e^{-\rho \mathcal{E}} (1 - e^{-\rho \mathcal{I}})]}{\rho}, \end{aligned} \tag{18}$$

yielding (4). The remaining two equations give the relations between \mathbf{e} , \mathbf{r} and \mathbf{i} . The first one yields

$$\begin{aligned} \mathbf{e} &= \lambda \mathbf{i} \int_0^{\infty} \mathbb{P}(\mathcal{E} > s) e^{-\rho s} ds \\ &= \frac{\lambda \mathbf{i}}{\rho} (1 - \mathbb{E} [e^{-\rho \mathcal{E}}]), \end{aligned} \tag{19}$$

and the third one gives, for $\rho > 0$,

$$\begin{aligned} \mathbf{r} &= \lambda \mathbf{i} \int_0^{\infty} \mathbb{P}(\mathcal{E} + \mathcal{I} \leq s) e^{-\rho s} ds \\ &= \frac{\lambda \mathbf{i}}{\rho} \mathbb{E} [e^{-\rho(\mathcal{E} + \mathcal{I})}]. \end{aligned}$$

Combining these with the constraint $\mathbf{e} + \mathbf{i} + \mathbf{r} = 1$ yields (5). Note that $\lambda \mathbf{i} = \rho$.

To prove the second part of the statement, we only need to check that the quantities defined in (6) satisfy

$$\begin{aligned} \mathbf{e}G_0^c(t) &= \lambda \mathbf{i} \int_{-\infty}^0 G^c(t-s)e^{\rho s} ds \\ \mathbf{i}F_1^c(t) + \mathbf{e}\Psi_0(t) &= \lambda \mathbf{i} \int_{-\infty}^0 \Psi(t-s)e^{\rho s} ds \\ \mathbf{r} + \mathbf{i}F_1(t) + \mathbf{e}\Phi_0(t) &= \lambda \mathbf{i} \int_{-\infty}^0 \Phi(t-s)e^{\rho s} ds. \end{aligned}$$

Indeed, combining these with (2) and the first part of the proposition, we obtain the fact that (3) solves (1). To check the first identity, write

$$\begin{aligned} \int_{-\infty}^0 G^c(t-s)e^{\rho s} ds &= \int_t^{+\infty} \mathbb{P}(\mathcal{E} > s)e^{\rho(t-s)} ds \\ &= \mathbb{E} \left[\mathbf{1}_{\{\mathcal{E} > t\}} \int_t^{\mathcal{E}} e^{\rho(t-s)} ds \right] \\ &= \frac{1}{\rho} \mathbb{E} \left[\mathbf{1}_{\{\mathcal{E} > t\}} (1 - e^{-\rho(\mathcal{E}-t)}) \right]. \end{aligned}$$

Multiplying by $\lambda \mathbf{i}/e$, we obtain

$$G_0^c(t) = \frac{\lambda \mathbf{i}}{e} \int_{-\infty}^0 G^c(t-s)e^{\rho s} ds = \frac{\mathbb{E} \left[\mathbf{1}_{\{\mathcal{E} > t\}} (1 - e^{-\rho(\mathcal{E}-t)}) \right]}{\mathbb{E} [1 - e^{-\rho \mathcal{E}}]}.$$

We conclude by noting that the term on right-hand-side equals $\mathbb{P}(\mathcal{E} - \Theta > t | \Theta < \mathcal{E})$, where Θ is an independent exponential random variable with parameter ρ .

Plugging (6) and (7) in the second equation, we obtain

$$\begin{aligned} \mathbf{i}F_1^c(t) + e\Psi_0(t) &= \mathbb{P}(\mathcal{E} \leq \Theta < \mathcal{E} + \mathcal{I}) \mathbb{P}(t + \Theta < \mathcal{E} + \mathcal{I} | \mathcal{E} \leq \Theta < \mathcal{E} + \mathcal{I}) \\ &\quad + \mathbb{P}(\Theta < \mathcal{E}) \mathbb{P}(\mathcal{E} \leq t + \Theta < \mathcal{E} + \mathcal{I} | \Theta < \mathcal{E}). \end{aligned}$$

Since, for $t \geq 0$,

$$\{t + \Theta < \mathcal{E} + \mathcal{I}\} \cap \{\mathcal{E} \leq \Theta < \mathcal{E} + \mathcal{I}\} = \{\mathcal{E} \leq t + \Theta < \mathcal{E} + \mathcal{I}\} \cap \{\mathcal{E} \leq \Theta < \mathcal{E} + \mathcal{I}\},$$

and

$$\{\mathcal{E} \leq t + \Theta < \mathcal{E} + \mathcal{I}\} \cap \{\mathcal{E} + \mathcal{I} \leq \Theta\} = \emptyset,$$

by the law of total probability,

$$\mathbf{i}F_1^c(t) + e\Psi_0(t) = \mathbb{P}(\mathcal{E} \leq t + \Theta < \mathcal{E} + \mathcal{I}).$$

On the other hand, since $\lambda \mathbf{i} = \rho$,

$$\begin{aligned} \lambda \mathbf{i} \int_{-\infty}^0 \Psi(t-s)e^{\rho s} ds &= \int_0^{+\infty} \mathbb{P}(\mathcal{E} \leq t + s < \mathcal{E} + \mathcal{I}) \rho e^{-\rho s} ds \\ &= \mathbb{P}(\mathcal{E} \leq t + \Theta < \mathcal{E} + \mathcal{I}). \end{aligned}$$

For the third equation, we proceed in the same way, plugging (6) and (7), we obtain

$$\begin{aligned} \mathbf{r} + \mathbf{i}F_1(t) + e\Phi_0(t) &= \mathbb{P}(\mathcal{E} + \mathcal{I} \leq \Theta) + \mathbb{P}(\mathcal{E} \leq \Theta < \mathcal{E} + \mathcal{I}) \mathbb{P}(\mathcal{E} + \mathcal{I} \leq t + \Theta | \mathcal{E} \leq \Theta < \mathcal{E} + \mathcal{I}) \\ &\quad + \mathbb{P}(\Theta < \mathcal{E}) \mathbb{P}(\mathcal{E} + \mathcal{I} \leq t + \Theta | \Theta < \mathcal{E}). \end{aligned}$$

Now, we note that, for $t \geq 0$,

$$\{\mathcal{E} + \mathcal{I} \leq t + \Theta\} \cap \{\mathcal{E} + \mathcal{I} \leq \Theta\} = \{\mathcal{E} + \mathcal{I} \leq \Theta\}.$$

As a result, applying the law of total probability, we obtain

$$\mathbf{r} + \mathbf{i}F_1(t) + \mathbf{e}\Phi_0(t) = \mathbb{P}(\mathcal{E} + \mathcal{I} \leq t + \Theta).$$

On the other hand, since $\lambda \mathbf{i} = \rho$,

$$\begin{aligned} \lambda \mathbf{i} \int_{-\infty}^0 \Phi(t-s)e^{\rho s} ds &= \int_0^{+\infty} \mathbb{P}(\mathcal{E} + \mathcal{I} \leq t+s) \rho e^{\rho s} ds \\ &= \mathbb{P}(\mathcal{E} + \mathcal{I} \leq t + \Theta). \end{aligned}$$

This concludes the proof of the first part of Proposition 2.

In the case $\rho < 0$, note that the computations in (18) and (19) remain valid, yielding the second part of the statement.

B Estimation procedure

Let us sum up the main steps of the estimation procedure proposed in the present work.

1. The first step is to define the set of dates d_1, d_2, \dots, d_k at which the contact rate $\lambda(t)$ changes in the population. This is done using external information (*e.g.* dates at which lockdowns come into force, dates at which preventive measures change, *etc.*). These dates correspond to times $t_1 = t_L, t_2, \dots, t_k$, where t_i is the time (in days) elapsed since the start of the epidemic at d_i . Since the time of the start of the epidemic is yet unknown, at this point one only knows the value of $t_{i+1} - t_i$ for $i \geq 1$.
2. We then estimate the growth rate of the epidemic ρ_i between each successive pair of dates of change of $\lambda(t)$, by fitting a set of exponential curves to the cumulative hospital data in each phase. Hence ρ_0 is the growth rate before d_1 , ρ_1 is the growth rate between d_1 and d_2 , and so on. The results of this step are depicted in Figure 3. This step also yields an estimate of the cumulative number of deaths at $t_1 = t_L$ (*i.e.* at the start of the first lockdown), $\Lambda_D(t_L)$, and the two constants A and B which appear in (16).
3. Using the growth rates ρ_i , $i \geq 0$, one then computes the corresponding effective contact rates $\lambda_{e,i}$, $i \geq 0$ using Proposition 2. Since we have assumed that $\bar{S}(t) \approx 1$ during the first two phases (*i.e.* up to t_2), we have $\lambda_0 = \lambda_{e,0}$ and $\lambda_1 = \lambda_{e,1}$.
4. The next step is to estimate the distribution of the delays between infection and hospital admission, ICU admission and hospital death, following the procedure of Subsection 1.7. The quantities ρ_0 , ρ_1 , A , B and $\Lambda_D(t_L)$ have been obtained in step 2 and λ_1 has been obtained in step 3. On the other hand, the constants C_E , C_I , C_R , defined in (10), depend only on δ , ρ_0 , ρ_1 and the distribution of $(\mathcal{E}, \mathcal{I})$, and can be computed easily by numerically solving the system of Definition 1. Using this, one can numerically invert (16) to find suitable parameters θ , k for each delay distribution. The result of this step is displayed in Figure 4, see also Table 4.
5. After this, one can also compute the infection-to-hospital admission ratio p_H defined in Subsection 1.8, using (17), and the corresponding ratio for ICU admission, both from the infection fatality ratio f which needs to be obtained from some external source.

6. The only unknown left at this point is the date of the start of the epidemic, d_0 , which is obtained from (15) and the estimated parameters of the delay distribution between infection and hospital death (recall that t_L is the time elapsed since the start of the epidemic at the start of lockdown measures, *i.e.*, between d_0 and d_1).
7. Finally, we solve the system of Definition 1 where the function $\lambda(t)$ is obtained in the following recursive way. We first set $\lambda(t) = \lambda_0$ for $t \leq t_L$, and solve the system up to $t_L = t_1$. After that, suppose that we have solved the system up to time t_i , corresponding to the date d_i , for some $i \geq 1$. We then let

$$\lambda(t) = \lambda_i := \frac{\lambda_{e,i}}{\bar{S}(t_i)}, \quad t_i \leq t < t_{i+1},$$

and we solve the system up to time t_{i+1} .

8. Once the trajectory of the process $(\bar{S}(t), \bar{E}(t), \bar{I}(t), \bar{R}(t))_{t \geq 0}$ has been computed, we can easily deduce the corresponding expected cumulative and daily number of hospital deaths $(\Lambda_D(t), t \geq 0)$, hospital admissions and ICU admissions, using (13). The result of this step is displayed in Figure 5.

C Inertia of the model with memory

One way to measure the inertia of the model is through the ratio

$$\frac{1 - \bar{S}(t_L + \delta)}{1 - \bar{S}(t_L)}, \quad (20)$$

where t_L is the time of the start of lockdown measures, and $\delta > 0$ is large enough that, at time $t_L + \delta$, the SEIR model is close to the exponential regime associated to the new growth rate, *i.e.* it satisfies (9) (in the present case, δ is of the order of a few weeks). Indeed, the longer it takes for daily new infections to start decreasing, the higher this ratio will be.

This ratio is displayed in Figure S1 for the model of Definition 1 where

$$(\mathcal{E}, \mathcal{I}) \sim \Gamma\left(k, \frac{m_E}{k}\right) \otimes \Gamma\left(k, \frac{m_I}{k}\right),$$

and $m_E = 2$ days and $m_I = 5$ days, for several values of $k \in (0, 10]$. In this way, $(\mathcal{E}, \mathcal{I})$ is a pair of independent Gamma random variables, with respective expectations m_E and m_I , and the parameter k controls the ratio between the variance and the expectation of these random variables. We observe on this figure that the inertia of the model seems to be an increasing function of the parameter k , *i.e.* it increases when the variance of \mathcal{E} and \mathcal{I} decreases. Note that the case $k = 1$ corresponds to $(\mathcal{E}, \mathcal{I})$ being a pair of exponential random variables, *i.e.* the Markovian ODE model.

While we do not assume that \mathcal{E} and \mathcal{I} follow Gamma distributions in this work, Figure S1 illustrates how the distribution of $(\mathcal{E}, \mathcal{I})$ affects the inertia of the model with memory of Definition 1.

Figure S1: Inertia of the model of Definition 1 around a change in the contact rate, measured by the ratio (20). For each value of k , we consider that \mathcal{E} and \mathcal{I} are independent Gamma random variables with parameters $(k, m_E/k)$ and $(k, m_I/k)$, respectively, with $m_E = 2$ and $m_I = 5$. The Markovian ODE model corresponds to $k = 1$ (red vertical line). We can see that, for this choice of distributions, decreasing the variance of $(\mathcal{E}, \mathcal{I})$ increases the inertia of the model with memory.

D Sensitivity analysis

We performed a Sobol sensitivity analysis using the SALib python package to investigate the effect of the distribution of $(\mathcal{E}, \mathcal{I})$ on the inferred value of R_0 . We let the mean exposed period vary between 2 and 5 days, the proportion of reported individuals between 0.2 and 0.8, and we let the mean infectious period of unreported individuals vary between 8 and 12 days. The resulting Sobol indices are given in Table S1. The parameters turned out to have very little interactions, with upper bounds for the confidence interval lying in the range 0.018-0.03.

Unsurprisingly, the parameter with the largest effect on the value of R_0 is the mean exposed period, as can already be seen in Corollary 3.

Parameter	First order Sobol indices
Mean exposed period	0.655 ± 0.019
Proportion of reported individuals	0.174 ± 0.012
Mean unreported infectious period	0.165 ± 0.011

Table S1: First order Sobol indices of R_0 relative to the three parameters.

E Change of the contact rate before the second lockdown

Figure S2 shows the result of the model predictions in the Grand Est and PACA regions when we assume a single change of the contact rate on October 30 at the start of the second lockdown. As we can see, the model grossly overestimates the peak during the second lockdown. This means that the contact rate must have been reduced slightly even before the start of the lockdown. This coincides with the time at which a nightly curfew was enforced in several large cities, including Marseilles and Strasbourg, the two largest cities in these regions. However, this does not mean that the curfew in itself is necessarily responsible for the reduction of the contact rate.

Figure S2: Model predictions of hospital admissions, ICU admissions and hospital deaths in the Grand Est and PACA regions without accounting for the (putative) effect of curfew before the second lockdown. To compute this, we assumed a single change of contact rate on October 30, with the rate computed from the halving time during the second lockdown.

Appendix C

Authors' Response to the Referee 1 of

“Estimating the state of the Covid–19 epidemic in France using a non–Markovian model”

We wish to thank the Referee for the great efforts and very helpful comments on our paper.

Answers to the minor comments:

- (1) Thanks for the information. We have added that paper in the Bibliography, and refer to it towards the end of the Introduction.
- (2) You raise an important point, which was already discussed in section 1.9, but certainly not well enough. We have introduced the notation λ_e which you suggest, as well as the effective R_e .
- (3) You are right : since both λ and the law of $(\mathcal{E}, \mathcal{I})$ determine the dynamics of the model, one expects that ρ is a given function of those data (that expectation is confirmed by (4), since the formula on the right is increasing in ρ). It turns out that we are rather interested in the inverse problem, which consists in evaluating the unknown parameter λ in terms of the easy to estimate parameter ρ , and the law of $(\mathcal{E}, \mathcal{I})$. We have added an explanation of that point just before the statement of Proposition 2.
- (4) It is true that the discussion around Figure 7 goes beyond the scope of the paper and that its practical implications are not clear. We have thus decided to remove this figure, along with its discussion. Note that the two curves were similar in the early stages of the epidemic because the two models were chosen so that they had the same initial growth rate.
- (5) Yes, you are right that some non-Markovian models have even less inertia than the Markovian ODE model. We discuss this in more detail in the Discussion and we added a figure in the Appendix to better illustrate our claim, *i.e.*, that considering more realistic distributions for $(\mathcal{E}, \mathcal{I})$ leads to more inertia in the model.

Appendix D

Authors' Response to the Referee 1 of

“Estimating the state of the Covid-19 epidemic in France using a non-Markovian model”

We wish to thank the Referee for the great efforts and very helpful comments on our paper.

Answers to the minor comments:

- (1) You are right that the model should include separate compartments corresponding to individuals who have been admitted to hospital, to ICU units, and those who have died in hospitals (note that we only count deaths that take place in hospitals), and that individuals in these compartments should not contribute to new infections (especially those who have died). However, such a model would take the same form as the one we consider, with the constraint that, on the event that an individual is admitted to hospital, we have $\mathcal{E} + \mathcal{I} < \mathcal{D}_H$ (and on the event that this individual dies in hospital, $\mathcal{D}_H < \mathcal{D}$, etc.). Since we did not use the joint distribution of $(\mathcal{E}, \mathcal{I})$ and \mathcal{D}_H or \mathcal{D} , we did not make this explicit in our manuscript. We have added a short discussion of this in Subsection 1.5.

In fact, the distributions we have chosen for $(\mathcal{E}, \mathcal{I})$ and \mathcal{D}_H (or \mathcal{D}) do not exactly satisfy this requirement (*i.e.* they cannot be coupled so that $\mathcal{E} + \mathcal{I} < \mathcal{D}_H$ on the event that the individual is admitted to hospital), since \mathcal{D}_H has support in $(0, \infty)$ while the support of $\mathcal{E} + \mathcal{I}$ is bounded away from 0. However, given that we find that $\mathbb{E}[\mathcal{D}_H] > 11$ days, these random variables can be coupled so as to make the probability that $\mathcal{D}_H < \mathcal{E} + \mathcal{I}$ fairly small. Multiplied by the probability that an individual is indeed admitted to a hospital (of the order of 2-3%), we argue that correcting this would not significantly affect the results.

Of course another option would be to assume that $\mathcal{D}_H = \mathcal{E} + \mathcal{I} + \mathcal{D}'$, where $\mathcal{D}' \geq 0$ is independent of $(\mathcal{E}, \mathcal{I})$, but this might not be realistic (individuals who end up in hospital are likely to have a different $(\mathcal{E}, \mathcal{I})$ distribution compared to those who don't), and the results would then also be harder to interpret, as we would obtain the mean delay between entry into the R compartment and hospital admission (or ICU admission / hospital death).

- (2) We have completely rewritten the beginning of section 1.1, and hope that now the introduction of the various r.v.'s is more clear. In particular, we have explained the necessity of defining different exposed/infectious (resp. infectious) periods for the initially exposed (resp. infectious) individuals, contrary to the case of the Markov models.
- (3) Thank you for this nice suggestion. We have included a concise step-by-step description of the estimation procedure in the appendix.